# Names Don't Matter:
# Symbol-Invariant Transformer for Open-Vocabulary Learning

**İlker Işık** [1]  **Wenchao Li** [1]

## Abstract

Current neural architectures lack a principled way to handle interchangeable tokens, i.e., symbols that are semantically equivalent yet distinguishable, such as bound variables. As a result, models trained on fixed vocabularies often struggle to generalize to unseen symbols, even when the underlying semantics remain unchanged. We propose a novel Transformer-based mechanism that is provably invariant to the renaming of interchangeable tokens. Our approach employs parallel embedding streams to isolate the contribution of each interchangeable token in the input, combined with an aggregated attention mechanism that enables structured information sharing across streams. Experimental results confirm the theoretical guarantees of our method and demonstrate substantial performance gains on open-vocabulary tasks that require generalization to novel symbols. Project page: `https://bu-depend-lab.github.io/Symbol-Invariant-Transformer/`

## 1. Introduction

Recent advances show that Transformer-based architectures can successfully tackle tasks once thought to require explicit symbolic manipulation, such as theorem proving (Han et al., 2022), mathematical reasoning (Rabe et al., 2021), and temporal logic solving (Hahn et al., 2021). These successes in language modeling over symbolic structures are supported by theoretical results showing that Transformers can represent the computation of any finite-state automaton (Liu et al., 2023). However, despite strong empirical performance and growing theoretical understanding, current neural architectures still lack a principled treatment of *open-vocabulary* reasoning, a core challenge in many symbolic domains.

A defining feature of symbolic systems is that many tokens are *interchangeable*: syntactically distinct yet semantically equivalent up to renaming. This arises naturally in logic, programming languages, and mathematics, where symbols act as placeholders rather than carriers of intrinsic meaning. In lambda calculus, for example, $\lambda x.x + 1$ and $\lambda y.y + 1$ are semantically identical; similarly, logic formulas differing only in variable names or atomic propositions are considered equivalent. This property, known as *alpha-equivalence*, means renaming interchangeable tokens should not affect meaning, yet neural models trained on fixed vocabularies tend to overfit to specific symbol identities and struggle with renamed variables (Orvalho & Kwiatkowska, 2026; Le et al., 2025; Wang et al., 2024).

Beyond robustness to renaming, a more fundamental challenge arises in open-vocabulary settings: when test-time symbols extend beyond the training vocabulary, fixed embedding tables have no representation for novel tokens. Standard approaches force a trade-off: embeddings must encode token identity to distinguish symbols, yet any reliance on identity both undermines invariance to renaming and blocks generalization to unseen tokens. This tension motivates the need for architectures that are *invariant by construction to permutations of interchangeable symbols*.

Linear Temporal Logic (LTL) (Pnueli, 1977), widely used in formal verification to specify temporal system properties, illustrates this challenge concretely: while Transformers can generate LTL witnesses and generalize to longer sequences (Hahn et al., 2021; Shiv & Quirk, 2019), generalization to larger or unseen proposition sets remains problematic, since atomic propositions are semantically interchangeable under renaming. LTL is one instance of a broader phenomenon; the same issue arises in any symbolic domain where alpha-equivalence plays a central role.

A recent line of work addresses this by constructing random embeddings for interchangeable tokens (Işık et al., 2025), enabling post-training vocabulary expansion and empirically improving robustness to renaming. However, the dependence on stochasticity for token discrimination provides no formal invariance guarantees (different random seeds can yield different predictions on alpha-equivalent inputs) and requires careful hyperparameter tuning.

---

[1]Department of Electrical & Computer Engineering, Boston University, USA. Correspondence to: İlker Işık <iilker@bu.edu>, Wenchao Li <wenchao@bu.edu>.

*Proceedings of the 43rd International Conference on Machine Learning*, Seoul, South Korea. PMLR 306, 2026. Copyright 2026 by the author(s).

We propose a **Symbol-Invariant Transformer** that achieves *exact invariance* to token renaming by design rather than by statistical encouragement. The architecture maintains parallel embedding streams, one per interchangeable token, processed by shared Transformer layers, yielding a provable guarantee that alpha-equivalent inputs always produce corresponding outputs.

We summarize our main contributions as follows:

- **Architecture**. We propose a novel symbol-invariant Transformer with parallel embedding streams (one per interchangeable token) combined through an aggregated attention mechanism. Our method introduces only a small number of additional hyperparameters and can be implemented as a lightweight modification to standard Transformer encoder–decoder architectures. It's even possible to convert an existing vanilla Transformer model into our architecture with minimal fine-tuning.

- **Theory**. Our architecture provides a formal guarantee of invariance to variable renaming: the model's outputs are identical for any pair of alpha-equivalent inputs, ensuring alpha-equivalence by construction.

- **Empirics**. We validate the approach on open-vocabulary symbolic reasoning tasks, demonstrating strong generalization and state-of-the-art performance, including outperforming GPT-5.2 on LTL witness generation.

## 2. Related Work

**Variable renaming.** Orvalho & Kwiatkowska (2026) demonstrate that large language model (LLM) performance on coding tasks degrades by up to 70% under semantics-preserving mutations, one of which is variable renaming—a phenomenon studied more extensively in prior work (Le et al., 2025; Barone et al., 2023; Wang et al., 2024). These studies consistently observe significant performance drops, suggesting that LLMs rely heavily on surface-level naming patterns rather than underlying semantics. This sensitivity to names is not confined to code: in natural language, the choice of proper nouns has been shown to affect accuracy on spatial reasoning tasks (Jha et al., 2022). Since the standard embedding tables inherently encode name-dependent biases by assigning distinct learned representations to distinct token identities, this vulnerability cannot be remedied by data augmentation or prompting alone; resolving it requires architectural changes that enforce invariance by construction.

**Symbol invariance & open-vocabulary learning.** Işık et al. (2025) introduced a metric for measuring robustness against variable renaming and proposed a novel embedding strategy using random vectors to handle interchangeable tokens, enabling post-training vocabulary expansion. However, as discussed previously, reliance on randomness leads to problems such as lack of theoretical guarantees. Ankner et al.

(2023) proposed a transformer architecture that is provably invariant to variable renaming without randomness, though vocabulary expansion was not considered in their work. Olsák et al. (2019) addressed symbol invariance in automated reasoning, but their focus was limited to graph neural networks and did not consider sequence-to-sequence tasks. Related work has also explored domain-specific vocabulary extensions using auxiliary information such as dictionary definitions or gesture components (Morazzoni et al., 2023; Wei et al., 2016), as well as general permutation invariance in transformers (Lee et al., 2019; Xu et al., 2024). While these papers tackle similar challenges, they do not address the specific problem of interchangeable tokens.

**Vocabulary generalization in other domains.** In computer vision, open-vocabulary image classification models are trained to recognize unseen classes at inference time (Radford et al., 2021; Wu et al., 2024; Tan et al., 2024). However, these methods are not applicable to interchangeable tokens as they rely on semantic relationships between seen and unseen classes derived from large-scale pre-training.

**Lexical generalization.** In natural language processing, lexical generalization has been studied to evaluate models' ability to generalize to novel words or phrases based on learned linguistic patterns (Kim & Linzen, 2020; Bandel et al., 2022; Kumon et al., 2024). Unlike our setting, lexical generalization emerges as a byproduct of training sequence-to-sequence models on large and diverse datasets, whereas post-training vocabulary expansion for interchangeable tokens requires explicit architecture and embedding design.

## 3. Preliminaries

### 3.1. Alpha-Equivalence

Many formal reasoning tasks involve interchangeable tokens such as bound variables or atomic propositions. Renaming these tokens does not change meaning. This property is known as *alpha-equivalence*. Formally, let the vocabulary be partitioned as $\mathbb{V} = \mathbb{V}_i \cup \mathbb{V}_n$, where $\mathbb{V}_i$ collects interchangeable tokens that can be renamed among themselves and $\mathbb{V}_n$ contains tokens with fixed identities. An *alpha-renaming* is any bijection $f: \mathbb{V} \to \mathbb{V}$ that leaves $\mathbb{V}_n$ unchanged and permutes $\mathbb{V}_i$; applying $f$ tokenwise to sequences yields alpha-converted input-output pairs. Two pairs are deemed *alpha-equivalent* if one can be obtained from the other via such a renaming function $f$. Since these pairs are semantically equivalent, a model that properly handles interchangeable tokens should produce consistent outputs across all such alpha-converted inputs.

Identifying $\mathbb{V}_i$ in practice is straightforward and domain-specific. For example, $\mathbb{V}_i$ corresponds to atomic propositions in logic, bound variables in programming languages, and universally quantified variables in theorem proving.

### 3.2. Problem Definition

The objective is to construct an embedding and representation scheme that is inherently robust to alpha-renaming while enabling post-training expansion of the interchangeable portion of the vocabulary. Formally, after training on $\mathbb{V} = \mathbb{V}_i \cup \mathbb{V}_n$, the model should operate on an augmented vocabulary $\mathbb{V}' = \mathbb{V}'_i \cup \mathbb{V}_n$ with $\mathbb{V}_i \subset \mathbb{V}'_i$, handling unseen interchangeable tokens at inference time. This setting motivates designs that preserve invariance across alpha-renamings yet maintain sufficient separability among interchangeable tokens to support accurate prediction.

### 3.3. Language Models and Transformers

The transformer architecture (Vaswani et al., 2017) revolutionized autoregressive language modeling by introducing a parallelizable attention mechanism that computes query, key, and value vectors from input embeddings to weigh token importance and capture long-range dependencies. The encoder-decoder transformers employ self-attention and cross-attention, use positional encodings to provide sequence structure, and apply attention masking during training to ensure causality. In this architecture, we employ three-way weight tying, whereby the final projection matrix and the embedding matrices of encoder and decoder are tied (Press & Wolf, 2016).

**Cosine loss.** Işık et al. (2025) apply feature normalization before the final projection to further improve learning. Given the output of the last layer $\boldsymbol{v}$ (the feature vector), instead of directly computing $\boldsymbol{U}\boldsymbol{v}$, we normalize it as $\boldsymbol{U}f_{fn}(\boldsymbol{v})$. Since $\boldsymbol{a} \cdot \boldsymbol{b} = \|\boldsymbol{a}\|\|\boldsymbol{b}\| \cos(\theta)$ where $\theta$ is the angle between $\boldsymbol{a}$ and $\boldsymbol{b}$, normalizing both the embeddings and the feature vector reduces the logits to cosine similarity, forcing the model to distinguish between tokens based solely on directions. This approach, known as cosine loss in the literature, has been successful in face recognition (Ranjan et al., 2017; Wang et al., 2017). The prior work employs the AdaCos loss function (Zhang et al., 2019) that scales the logits adaptively throughout training, adapting it for sequence modeling by treating sequence length as a batch dimension. We also use this technique in our experiments.

## 4. Proposed Method

We present a novel neural architecture for handling interchangeable tokens – semantically equivalent symbols that are distinguishable from each other, such as variable names or atomic propositions (see Figure 1). Our method maintains $k$ parallel embedding streams, one for each interchangeable token, enabling the model to process each token's context independently while sharing information across streams. The approach combines three key ideas: (1) parallel embedding with actual and placeholder representations, (2) dual

self-attention over individual streams and aggregated views, and (3) projection that averages non-interchangeable token logits while preserving interchangeable token distinctions. Details are provided in Appendix A.

### 4.1. Creation of Streams and Per-Stream Operations

For each of the $k$ interchangeable tokens, we create a parallel embedding stream by replacing: (1) positions containing that token with an *actual* embedding index, (2) positions containing other interchangeable tokens with a *placeholder* embedding index, and (3) base tokens remain unchanged. This yields $k$ parallel embeddings that capture different "views" of the input, each centered on a different interchangeable token. A binary mask tracks which positions contain each token for later use. Details appear in Algorithm 1.

Within each stream, we apply per-stream self-attention to allow independent context building, followed by a standard feed-forward network. Layer normalization and residual connections follow standard Transformer design. These per-stream operations are parameterized identically across all streams, i.e., all $k$ streams share the same weights for attention and feed-forward layers. This parameter sharing enables post-training vocabulary extension, where new interchangeable tokens can be added without retraining.

### 4.2. Aggregation and Projection

To enable information sharing across streams, we aggregate them into a single view by averaging hidden states across all $k$ streams, then restoring the true hidden state from stream $i$ at positions where interchangeable token $i$ appears. This approach combines distributed context from all streams while preserving the specialized representations at interchangeable token positions. The projection step uses a similar aggregation: base tokens are averaged across all $k$ streams (equally valid from all perspectives), while interchangeable token $i$'s logit comes from stream $i$ (its specialized stream). Algorithms 2 and 5 provides details for these steps.

### 4.3. Overall Architecture

The encoder layer is composed of per-stream self-attention, aggregated attention, and feed-forward operations (steps 2, 3, and 5 in Figure 1). The decoder extends the encoder with causal masking and cross-attention to encoder outputs. Like the encoder, it applies per-stream self-attention followed by aggregated-view attention. The cross-attention can operate in either per-stream mode (each decoder stream attends to its corresponding encoder stream) or aggregated mode (all streams attend to aggregated encoder representations), which correspond to 4A and 4B in Figure 1. Algorithms 3 and 4 detail these layers. Note that this architecture extends naturally to decoder-only autoregressive models with the

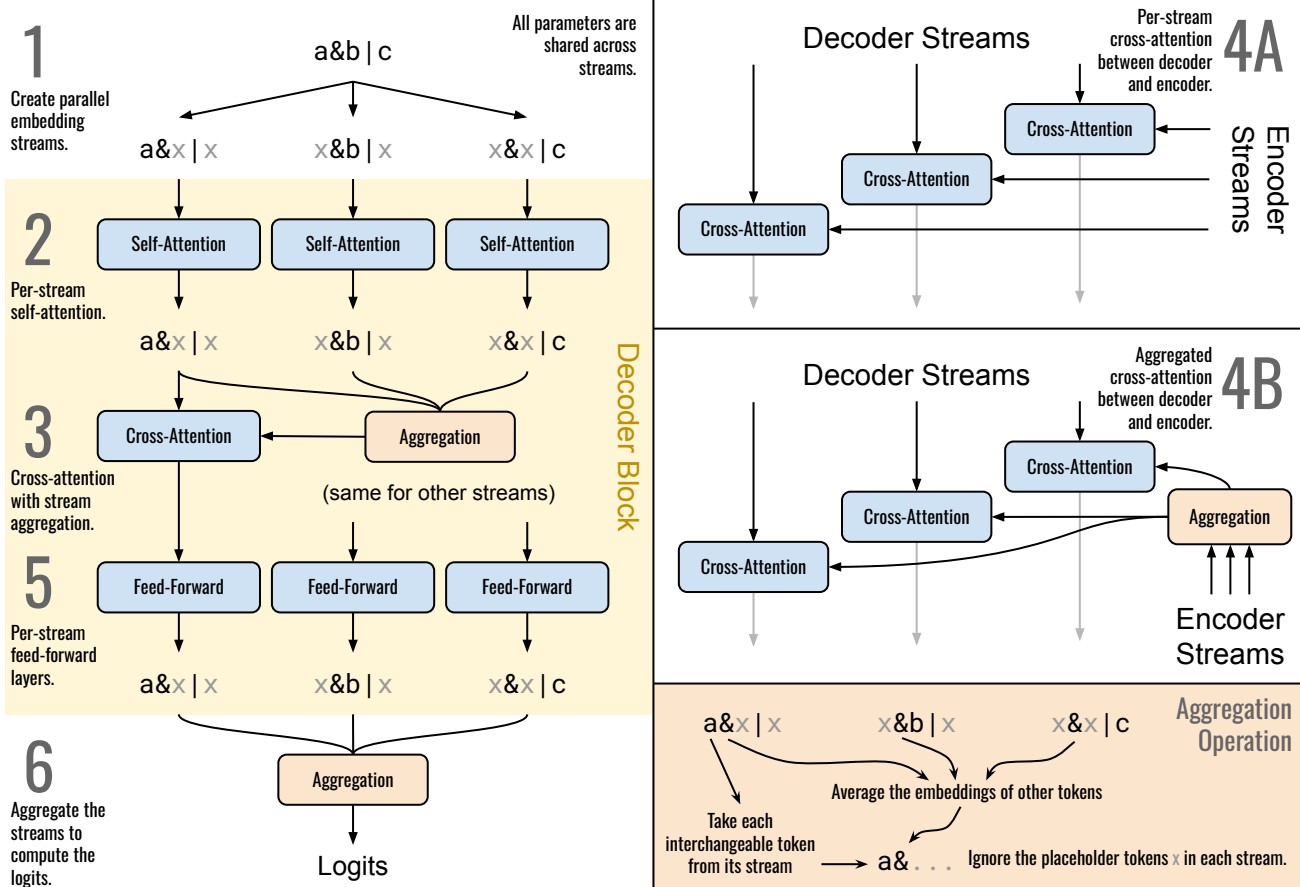

*Figure 1.* Method overview. The numbered sections correspond to: (1) Creation of parallel embedding streams with actual and placeholder embeddings for interchangeable tokens, (2) per-stream self-attention over individual streams, (3) aggregated attention between each stream and the fused view, (4A/4B) cross-attention between decoder streams and encoder streams with per-stream or aggregated modes, (5) applying feed-forward networks per stream, and (6) projection that averages base token logits and preserves interchangeable token logits. The bottom right section illustrates the aggregation process that fuses parallel streams into a single view while restoring true embeddings at interchangeable token positions. All parameters are shared across streams, which enables post-training vocabulary extension. Common transformer components (e.g., layer normalization, residual connections) are omitted for clarity.

removal of cross-attention to encoder outputs.

### 4.4. Theoretical Guarantees

We now formally establish that the proposed architecture guarantees alpha-equivalence by construction.

**Theorem 4.1** (Alpha-Renaming Invariance). *Let $M$ denote the proposed model and $f : \mathbb{V} \to \mathbb{V}$ be an alpha-renaming function that fixes $\mathbb{V}_n$ and permutes $\mathbb{V}_i$. For any input sequence $x$, let $\hat{y} = M(x)$ denote the model's output. Then for the alpha-renamed input $x' = f(x)$ obtained by applying $f$ tokenwise, the model produces output $\hat{y}' = M(x')$ such that*

$$f^{-1}(\hat{y}') = \hat{y}. \tag{1}$$

*Equivalently, $\hat{y}' = f(\hat{y})$, i.e., the model's prediction on alpha-renamed input is the alpha-renamed version of the original prediction.*

**Intuition.** The key insight is that alpha-renaming merely permutes the order of the $k$ parallel streams without altering the computations within or across them. Specifically, if alpha-renaming $f$ maps interchangeable token $i$ to position $j$ in the vocabulary, then stream $i$ in the original computation corresponds exactly to stream $j$ in the renamed computation. All operations in our architecture fall into two categories: *per-stream operations* (self-attention, feed-forward networks, layer normalization) and *aggregated operations* (averaging hidden states across streams). Per-stream operations are unaffected by stream reordering because each stream processes its input independently using shared parameters. Aggregated operations remain unchanged because they rely on extraction of each interchangeable token from its corresponding stream (which is unaffected by permutation) and summation, which is commutative: $\sum_{i=1}^{k} v_i = \sum_{i=1}^{k} v_{\pi(i)}$ for any permutation $\pi$. The full proof is provided in Appendix B.

# 5. Experiments

## 5.1. Tasks and Evaluation Metrics

Following prior work (Işık et al., 2025), we evaluate the proposed method across three distinct tasks that test vocabulary generalization capabilities in different contexts.

**Copying with scalable vocabulary.** Copying sequences has been used as a toy task in the literature to evaluate sequence modeling capabilities (Su et al., 2024), especially for assessing the generalization capability to longer sequences in the context of positional encodings. The prior random embedding method (Işık et al., 2025) formulated a new variant of the copying task to evaluate vocabulary generalization jointly with sequence length generalization. The experimental procedure involves testing the model on sequences that contain tokens outside the training vocabulary, in addition to being longer than the training sequences. The results are presented in Appendix E.

**Propositional logic assignment prediction.** This task requires generating satisfying assignments for propositional formulae composed of atomic propositions and logical operators including negation, conjunction, disjunction, equivalence, and exclusive-or (see Appendix C.1 for formal definitions). Given a formula, the model must predict a sequence of atomic propositions (APs) and their Boolean values that collectively satisfy the formula. The task permits partial assignments where only a subset of propositions receives explicit values. Our evaluation is based on PropRandom35 from DeepLTL (Hahn et al., 2021), and we leverage `pyaiger` (Vazquez-Chanlatte & Rabe) to validate the correctness of predicted assignments.

**LTL (Linear Temporal Logic) witness generation.** This task extends propositional logic with temporal operators expressing properties over time (see Appendix C.2 for formal definitions). Given an LTL formula, the model must generate a symbolic trace consisting of a finite prefix followed by a repeating suffix that satisfies the formula's temporal constraints. Although LTL is technically a superset of propositional logic, the propositional logic variant studied here incorporates derived operators (equivalence and exclusive-or) that are absent in our LTL formulation (for consistency with prior work). Consequently, these tasks pose distinct reasoning challenges: propositional logic emphasizes relational reasoning between APs through derived operators, while LTL primarily demands temporal reasoning. Our evaluation is based on LTLRandom35 from DeepLTL (Hahn et al., 2021), and model predictions are validated using `spot 2.11.6` (Duret-Lutz et al., 2022).

### 5.1.1. ALPHA-COVARIANCE

To measure the degree of robustness against alpha-renaming, we employ the alpha-covariance metric introduced by Işık

et al. (2025). As defined in Section 3.1, alpha-equivalent pairs can be generated through systematic renamings of interchangeable tokens. Given $V_i = |\mathbb{V}_i|$ interchangeable tokens and an input-output pair $(\boldsymbol{x}, \boldsymbol{y})$ containing $k$ distinct occurrences from $\mathbb{V}_i$, there exist $P(V_i, k) = V_i!/(V_i - k)!$ alpha-equivalent pairs generated through permutations.

The alpha-covariance metric quantifies consistency across these alpha-equivalent inputs. Let $\mathbb{P} = \{(\boldsymbol{x}^1, \boldsymbol{y}^1), \ldots, (\boldsymbol{x}^n, \boldsymbol{y}^n)\}$ be $n$ input-output pairs alpha-equivalent to $(\boldsymbol{x}, \boldsymbol{y})$, with $\alpha_i$ denoting the corresponding alpha-renaming satisfying $\alpha_i(\boldsymbol{x}) = \boldsymbol{x}^i$ and $\alpha_i(\boldsymbol{y}) = \boldsymbol{y}^i$. For each input $\boldsymbol{x}^i \in \mathbb{P}$, the model generates prediction $\hat{\boldsymbol{y}}^i$. Define $\mathbb{U} = \{\alpha_i^{-1}(\hat{\boldsymbol{y}}^i) \mid 1 \leq i \leq n\}$ as the set of predictions with alpha-conversions undone. Ideally, $|\mathbb{U}| = 1$ since all predictions should collapse to the same output when conversions are reversed. The alpha-covariance is computed as:

$$1 - \frac{|\mathbb{U}| - 1}{|\mathbb{P}| - 1} \tag{2}$$

An alpha-covariance of 1 indicates perfect invariance to alpha-conversions, while 0 indicates maximum sensitivity (every prediction differs after undoing conversions).

## 5.2. Experimental Setup and Baselines

**Architecture.** We employ a transformer-based encoder-decoder architecture throughout our evaluation suite. Weight sharing is applied between encoder and decoder embeddings to maintain consistency in representation dimensionality. Complete hyperparameter specifications are documented in Table 5 located in Appendix D. RoPE (Su et al., 2024) serves as the default positional mechanism across all experiments. In the logic-based tasks, tree-positional encoding is utilized in the encoder, and beam search with beam width $k = 3$ is applied during generation.

**Baseline methods.** We compare our method against three baselines. The **full vocabulary baseline** sidesteps the vocabulary generalization challenge by utilizing fixed learned embeddings trained on a training set with expanded vocabulary that matches the test set size. This baseline represents an upper bound on the performance of a standard transformer model achievable through vocabulary expansion via training data augmentation. The **alpha-renaming baseline** applies fixed learned embeddings with an expanded vocabulary that matches the test set size on the original training set. The **random embedding baseline** employs the dual-part random embedding approach from prior work (Işık et al., 2025), which uses embeddings with two components: a shared learnable part to convey semantic equivalence and a randomly-generated part to distinguish between interchangeable tokens. In each task, we use the best hyperparameter configuration identified in the prior work.

*Table 1.* Evaluation of baselines and the proposed method under different perturbations. The alpha-renaming baseline uses 5 AP embeddings since vocabulary generalization is not evaluated here. Columns show: (1) task type, (2-3) training dataset and model type, (4-5) correct predictions and exact matches to the ground truth labels on the test set, and (6-8) mean alpha-covariance across varying AP counts, computed on all alpha-equivalent permutations of 1000 test samples (100 for GPT-5.2).

| Task | Training Dataset | Model | Evaluation | | Alpha-Covariance | | |
|---|---|---|---|---|---|---|---|
| | | | Correct | Exact | 3 AP | 4 AP | 5 AP |
| Propositional Logic | Normal | Baseline | 95.62% | 57.94% | 95.70% | 93.69% | 76.02% |
| | Renamed | Baseline | 41.57% | 9.04% | 14.96% | 16.85% | 10.65% |
| | | Alpha-Renaming | 93.85% | 57.24% | 99.56% | 99.60% | 93.23% |
| | | Random Embedding | 93.25% | 56.45% | 99.23% | 99.42% | 92.98% |
| | | Proposed | **98.03%** | **60.96%** | **100.0%** | **100.0%** | **100.0%** |
| | Reduced | Baseline | 63.26% | 29.31% | 86.21% | 75.07% | 53.31% |
| | | Alpha-Renaming | 57.48% | 26.77% | 97.89% | 97.35% | 95.06% |
| | | Random Embedding | 55.23% | 26.28% | 97.88% | 95.79% | 90.18% |
| | | Proposed | **70.43%** | **35.81%** | **100.0%** | **100.0%** | **100.0%** |
| | Pretrained | GPT 5.2 | 99.73% | 25.60% | 42.97% | 29.87% | 1.03% |
| LTL | Normal | Baseline | 98.23% | 83.23% | 96.87% | 95.86% | 91.80% |
| | Renamed | Baseline | 34.13% | 12.12% | 64.93% | 57.99% | 40.91% |
| | | Alpha-Renaming | 97.96% | 77.66% | 99.55% | 99.49% | 98.86% |
| | | Random Embedding | 95.94% | 76.45% | 97.66% | 97.76% | 98.29% |
| | | Proposed | **98.24%** | **79.65%** | **100.0%** | **100.0%** | **100.0%** |
| | Reduced | Baseline | 87.47% | 63.61% | 94.37% | 91.70% | 85.64% |
| | | Alpha-Renaming | 89.50% | 64.15% | 99.02% | 98.67% | 97.82% |
| | | Random Embedding | 87.32% | 59.04% | 97.94% | 96.12% | 94.34% |
| | | Proposed | **93.46%** | **68.63%** | **100.0%** | **100.0%** | **100.0%** |
| | Pretrained | GPT 5.2 | 86.83% | 35.93% | 81.76% | 82.97% | 77.56% |

*Table 2.* Ablation results for the propositional logic task. The first model is the best performing configuration, other models either add (+) or remove (-) components. The accuracy is reported on the heatmap test set.

| Best Model | 95.05% |
|---|---|
| +CA | 92.66% |
| -CP+CA | 28.51% |
| -EA | 92.47% |
| -DA | 84.48% |
| -EA-DA | 72.35% |
| -EP | 91.44% |
| -DP | 46.55% |

*Table 3.* LTL ablation results.

| Best Model | 90.47% |
|---|---|
| +CA | 90.27% |
| -CP+CA | 20.93% |
| -EA | 84.13% |
| +DA | 89.47% |
| -EP | 84.13% |
| -EP-DP+DA | 20.27% |

## 5.3. Dataset Perturbations

In this experiment, the inductive bias of our method and the baselines is assessed by introducing controlled perturbations to the benchmark datasets. The first type of perturbation involves renaming the atomic propositions (APs) in the inputs and outputs such that the order of the first AP appearances in the trace is always the same. Although this modification does not alter the logical structure of the formulas, the models without favorable inductive bias may depend on AP order to make predictions, leading to performance degradation. The second perturbation type reduces the sample count in the training dataset from 800K to 80K, thereby limiting the exposure of models to diverse AP combinations and formula structures during training. A helpful inductive bias should enable the model to perform well even with reduced training data, improving sample efficiency.

The results in Table 1 demonstrate that the proposed method consistently outperforms all baselines in each perturbation scenario across both tasks. Superior performance on the renamed dataset indicates that the proposed method effectively captures the underlying logical structure of the formulas. Interestingly, the proposed method trained on the renamed dataset even surpasses the baseline trained on the

unmodified dataset, while other baselines that enable vocabulary extension do not. This can be interpreted as a pareto improvement in the bias-variance trade-off (better in-distribution and out-of-distribution performance). Similarly, the results on the reduced dataset highlight the superior sample efficiency of the proposed method. The proposed method achieves perfect alpha-covariance across all AP counts in both training regimes, validating Theorem 4.1.

## 5.4. Vocabulary & Length Generalization

From a computational standpoint, expanding training coverage by enumerating more atomic propositions (APs) and longer formulas rapidly becomes impractical. The time required to synthesize datasets and validate traces grows steeply – empirically trending toward exponential in both AP count and formula length, as shown in previous work (Işık et al., 2025). Consequently, approaches that generalize to larger AP sets and longer sequences without retraining on matching datasets substantially reduce dataset generation and training costs. This, combined with the PSPACE-complete complexity of the LTL task, makes such generalization a key driver of practical scalability.

In this section, we assess the performance when confronted

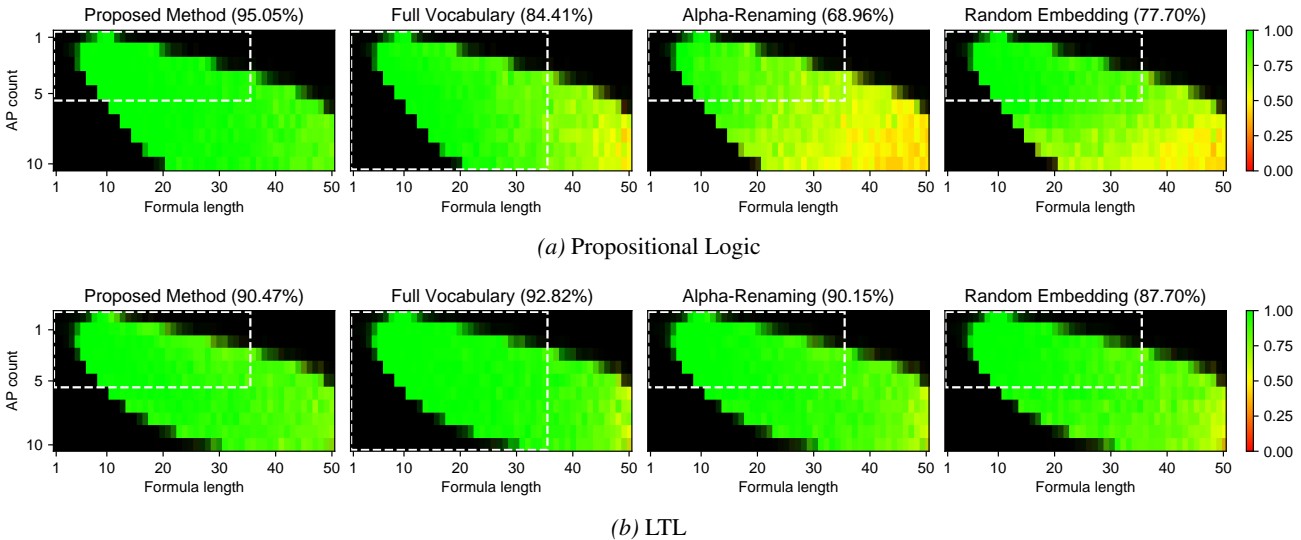

*Figure 2.* Heatmaps showing prediction accuracy on test sets with varying formula complexity for LTL (top) and propositional logic (bottom) tasks. Brightness indicates sample density, with full brightness representing 100 samples. The dashed white box marks the training data distribution boundary.

with test distributions that extend beyond training coverage. This experiment uses the test set from prior work (Işık et al., 2025), which systematically varies both the number of APs and formula lengths to create a grid of out-of-distribution scenarios. Each combination of AP count and formula length contains at most 100 test samples. The maximum formula length in this test set is 50, exceeding the training maximum of 35. Similarly, the maximum AP count in the test set is 10, surpassing the training maximum of 5. Although the maximum AP count and the formula length are increased, the probability distribution of operators remains consistent with the training set. This ensures that the core reasoning task remains unchanged, isolating the evaluation to vocabulary and length generalization. For the LTL task, we identify and address an imbalance in the training set. The details are documented in Appendix F.

Figure 2 presents heatmaps illustrating model accuracy across varying levels of formula complexity for both tasks. The propositional logic task reveals substantial performance differences under out-of-distribution generalization, with the proposed method markedly outperforming all baselines, even the full vocabulary approach. This demonstrates the effectiveness of our approach in capturing the underlying logical structure. In the LTL task, however, the results are more closely clustered, with all methods achieving similar accuracies. These performance patterns stem from fundamental differences in operator distributions and reasoning requirements. The propositional logic task contains relational operators (implication, exclusive-or) that require the model to reason about AP relationships, whereas the LTL task relies on simpler conjunction and disjunction operators in addition to temporal operators. Consequently, the vocab-

ulary generalization challenge is less pronounced in LTL – all models perform similarly because temporal reasoning, not AP relationship handling, is the primary bottleneck. In propositional logic, modeling inter-AP relationships constitutes the major challenge, and this is where the proposed method's inductive bias towards permutation invariance provides the greatest advantage. The dual-part embedding design in random embeddings (Işık et al., 2025) also benefits from this distinction: it forces the model to separate common and distinguishable aspects of APs, proving beneficial when inter-proposition relationships dominate.

### 5.5. Ablation Analysis

We ablate various attention components to assess their contributions to overall performance. We use two letter codes to denote each attention mechanism. The first letter indicates the block: 'E' for encoder, 'D' for decoder, and 'C' for cross-attention. The second letter indicates the attention type: 'P' for per-stream and 'A' for aggregated attention.

For each task, we determined the best configuration by hyperparameter tuning on a separate validation set with 10 APs. The best configuration is EP-DP-EA-DA-CP for propositional logic and EP-DP-EA-CP for LTL. These configurations form the baseline for ablation studies. Tables 2 and 3 summarize the ablation results. For full details on each ablation model and heatmaps, please refer to Appendix G.

**Cross-attention.** A critical finding emerges from comparing per-stream cross-attention (CP) with aggregated cross-attention (CA). Replacing per-stream cross-attention with aggregated cross-attention results in catastrophic performance degradation, with accuracies plummeting to 20.93%

for LTL and 28.51% for propositional logic. This dramatic loss becomes increasingly pronounced as the number of APs increases, empirically demonstrating that without per-stream cross-attention, the decoder streams cannot distinguish which encoder stream they correspond to. Conversely, adding aggregated cross-attention in addition to per-stream cross-attention results in marginal decreases.

**Per-stream attention.** Removing DP leads to significant accuracy drops in both tasks, while removing EP has a smaller impact. This aligns with the observation that per-stream attention is vital for stream identification, which also explains why DP is more critical since the decoder must accurately identify streams to generate correct outputs.

**Aggregated attention.** Removing DA results in a significant accuracy drop to 84.48%, while removing EA has a minor impact, with accuracy declining only to 92.47%. This asymmetry suggests that relational understanding in the decoder is crucial for the propositional logic task. Note that even when EA is removed, DA still offers an indirect computation path for relating AP tokens originating from the encoder. When both aggregated attention mechanisms are removed, accuracy drops substantially to 72.35%, which underlines the importance of aggregated attention for relational reasoning in this task. Unlike propositional logic, LTL task benefits from the removal of DA, with accuracy slightly increasing from 89.47% to 90.27%. The removal of EA in addition to DA leaves the model without any aggregated attention, yet the accuracy drop to 84.13% is not as severe as in the propositional logic case. This confirms that relational reasoning (which aggregated attention primarily supports) is not the critical bottleneck for the LTL task.

**Overall insights.** These ablation results demonstrate that both per-stream and aggregated attention mechanisms play essential and complementary roles in the proposed architecture since the best configuration in each task includes both. However, where these mechanisms should be activated depends on the task. Per-stream attention is universally critical for stream identification, while aggregated attention's importance varies by task according to the prominence of relational reasoning requirements. While the best configuration should be determined through hyperparameter tuning for each task, we recommend the configuration EP-DP-EA-DA-CP as a strong default choice.

### 5.6. Evaluation Against GPT-5.2

We benchmark our method against GPT-5.2 (Singh et al., 2025), a state-of-the-art general-purpose large language model, to contextualize the results. GPT-5.2 was configured with medium reasoning effort and prompted with four few-shot examples per task, covering a range of formula complexities and demonstrating the expected input/output format (Listings 1 and 2, Appendix H). For the propositional

logic task, model outputs were additionally constrained to JSON format via schema validation to minimize formatting errors. Full prompt details are documented in Appendix H.

We limit the sample count per cell to 10 in the heatmap test set due to financial constraints. On propositional logic, GPT-5.2 achieves 99.49% accuracy. However, on LTL, it achieves only 81.45% accuracy, substantially underperforming all other methods, which underscores the value of specialized models tailored to specific domains. Despite superior performance on propositional logic, GPT-5.2's practical utility is constrained by computational overhead. With medium reasoning effort, GPT-5.2 requires 10–90 seconds per sample, averaging 37 seconds on 387 LTL test samples. In contrast, our proposed method generates predictions nearly instantaneously on consumer-grade hardware.

**Top-N accuracy.** To match GPT-5.2's performance on propositional logic, we consider the top-N accuracy of our method. Our model is on par with GPT-5.2 at top-10 and top-25 accuracy, achieving 99.03% and 99.54% respectively. More details are provided in Appendix I.

### 5.7. Computational Cost Analysis

A standard attention mechanism has a time complexity of $O(L^2)$, where $L$ is the sequence length. With $S$ streams, our architecture scales to $O(SL^2)$. If $S$ were extremely large, the added complexity could make the method impractical. In practice, however, $S$ is typically much smaller than $L$, and empirically, the method remains practical for $S = 10$: average time per sample increases from 3.38 ms to 5.13 ms on propositional logic (Figure 9 in Appendix J). Computing top-25 samples instead of top-3 quadruples the time, which is still instantaneous compared to GPT-5.2.

**Memory usage.** Total activation memory grows as $O(SLd)$ since each stream maintains a separate hidden state tensor of shape $L \times d$. On the LTL heatmap, the memory usage from 5 to 10 APs scales from 1.9 GB to 3.9 GB for the proposed method, and from 3.1 GB to 3.2 GB for the full vocabulary baseline (for a batch size of 64). Note that the proposed model uses less parameters (Table 5) since we observed better parameter efficiency during preliminary hyperparameter tuning, alleviating the memory scaling problem.

### 5.8. Converting & Fine-Tuning Pre-Trained Models

A natural question is whether the proposed method can be applied to an existing pre-trained model rather than training from scratch. When all aggregated attention components are disabled, the architecture introduces no additional parameters compared to a standard transformer baseline, making a parameter-preserving conversion straightforward. Specifically, given a pre-trained baseline model trained on a fixed interchangeable token vocabulary, we perform the following

*Table 4.* Model conversion & fine-tuning results on the LTL and propositional logic tasks. The last row reports the EP-DP-CP model trained from scratch (see Appendix G). The pre-trained baseline uses fixed embeddings trained on 5 APs and cannot generalize to larger vocabularies. The converted models are initialized from the pre-trained baseline and fine-tuned using the procedure described in Section 5.8. Heatmap column denotes the mean accuracy over the full out-of-distribution heatmap test set.

| Model | Propositional Logic | | | | LTL | | | |
|---|---|---|---|---|---|---|---|---|
| | 5 AP Val | 10 AP Val | Heatmap | Fig. | 5 AP Val | 10 AP Val | Heatmap | Fig. |
| Pre-trained baseline | 95.57% | — | — | | 97.96% | — | — | |
| Converted & fine-tuned (1 epoch) | 87.54% | 67.17% | 70.11% | 10a | 94.75% | 92.44% | 85.91% | 11a |
| Converted & fine-tuned (5 epochs) | 90.55% | 68.78% | 71.35% | 10b | 95.12% | 92.02% | 85.88% | 11b |
| Trained from scratch | 91.47% | 69.97% | 72.35% | 5i | 94.77% | 91.34% | 84.13% | 6i |

conversion: (1) use the embedding of one interchangeable token (e.g., "a") as the actual embedding (Section 4.1); (2) use the embedding of another interchangeable token (e.g., "b") as the placeholder embedding; (3) map all remaining parameters one-to-one, then fine-tune.

Table 4 presents the results on both tasks. Full heatmaps for the fine-tuned models are provided in Appendix K. On LTL, the converted and fine-tuned models achieve accuracy comparable to the model trained from scratch, and even outperform it on the out-of-distribution heatmap test set after only one fine-tuning epoch. On propositional logic, fine-tuning similarly closes the gap with the from-scratch baseline within a small number of epochs. This demonstrates that the method can be effectively initialized from an existing pre-trained model: the embedding conversion requires no architectural changes, and a small number of fine-tuning steps is sufficient to recover, and in some metrics even surpass, from-scratch performance. The pre-trained baseline cannot process vocabularies larger than those seen during training, so its out-of-distribution columns are not applicable. This suggests a practical path for extending pre-trained models with alpha-invariance guarantees without discarding prior training.

## 6. Limitations

The computational cost of the proposed method scales as $O(SL^2)$ with the number of interchangeable token streams $S$, which imposes a practical upper bound on vocabulary size. Although the experiments demonstrate tractability for $S \leq 10$ (Section 5.7), domains such as program synthesis or theorem proving may involve hundreds of distinct local variables, where this cost becomes prohibitive. We use the term *open-vocabulary* to describe generalization to interchangeable token sets strictly larger than those seen during training, not an unbounded vocabulary in practical sense, since $S$ is constrained by available compute and memory. Sparsification strategies could mitigate this cost, but they must be designed carefully to preserve the alpha-equivalence guarantee: approaches that drop streams based on token identity would break invariance, whereas selection based

on input-symmetric criteria (such as positional frequency) could potentially preserve it. Although the memory usage was well within the hardware capacity ($\leq 4$ GB, Section 5.7) for our LTL setting ($S = 10$, $L = 50$, $d = 64$), it could become a bottleneck for substantially larger $S$ or $d$. The same sparsification strategies that mitigate time complexity (e.g., Top-K stream gating) would apply equally here.

A second limitation concerns the generation of novel interchangeable tokens. The proposed method can produce output tokens that do not appear in the input, as in a conventional language model, but any output symbol that must be treated as an interchangeable token requires a corresponding embedding stream. Because streams are instantiated from the encoder input, the model cannot generate a fresh interchangeable token, i.e., one with no corresponding stream. For the tasks studied here, namely satisfying-assignment prediction and LTL witness generation, all output interchangeable tokens are drawn from the input formula, so this constraint does not affect the reported benchmarks. Tasks that require inventing new variable names, such as constructive proof generation or code synthesis, are therefore outside the current scope; a natural extension is to maintain a reserved pool of fresh-symbol streams for such settings. The model conversion & fine-tuning procedure introduced in Section 5.8 further suggests that it may be possible to address these limitations incrementally, opening pathways toward applications in coding and mathematical reasoning.

## 7. Conclusion

This work demonstrates that symbol-invariant attention yields strong open-vocabulary generalization, delivering consistent gains over random embeddings and standard baselines on both propositional logic and LTL tasks. The method is alpha-invariant by construction, achieving perfect alpha-covariance across perturbations and AP counts. The experiments show that the inductive bias is especially beneficial when relational reasoning dominates, while remaining competitive when temporal reasoning is the primary bottleneck. Using a simple procedure, our method can also be retrofitted into existing pre-trained models with minimal fine-tuning.

## Acknowledgement

The authors thank the anonymous reviewers for their constructive feedback and suggestions. This work was supported in part by the U.S. National Science Foundation under grant CCF-2340776.

## Impact Statement

This paper presents work whose goal is to advance the field of Machine Learning. There are many potential societal consequences of our work, none of which we feel must be specifically highlighted here.

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

# A. Algorithm Details

This appendix provides detailed pseudocode for all algorithms used in the interchangeable token architecture. The main paper describes the intuition and design of each component; readers interested in implementation details may refer to the pseudocode below.

**Embedding with Interchangeable Tokens.**   The embedding algorithm processes the input token sequence to create $k$ parallel embeddings, one for each interchangeable token. For each stream $i$, the algorithm modifies the token sequence to replace the interchangeable token $i$ with an actual embedding index and all other interchangeable tokens with a placeholder embedding index. A binary mask is simultaneously created to track where each interchangeable token appears in the input, which will later be used for masking predictions. See Algorithm 1.

**Aggregating Interchangeable Token Streams.**   The aggregation algorithm fuses the $k$ parallel streams into a single aggregated stream while preserving the true representations at interchangeable token positions. It first computes a mean representation by averaging hidden states across all streams, then selectively restores the actual hidden state from stream $i$ at positions where interchangeable token $i$ appears. See Algorithm 2.

**Transformer Encoder Layer.**   The encoder layer implements dual attention over the parallel streams: first, each stream applies self-attention independently to build context, then each stream attends to an aggregated view of all streams' representations. This dual mechanism enables both stream-specific processing and cross-stream information sharing. A standard feed-forward network is applied after each attention operation, with layer normalization and residual connections throughout. See Algorithm 3.

**Transformer Decoder Layer.**   The decoder layer mirrors the encoder but with causal masking for auto-regressive generation. Following per-stream self-attention under a look-ahead mask, the decoder applies aggregated-view attention as in the encoder. Cross-attention then allows each decoder stream to condition on encoder outputs, with configurable modes: per-stream cross-attention where each decoder stream attends to its corresponding encoder stream, or aggregated cross-attention where each decoder stream attends to an aggregated encoder representation. Multiple cross-attention layers can be stacked with different modes. See Algorithm 4.

**Projection.**   The projection algorithm converts the parallel hidden states back to vocabulary logits using an asymmetric aggregation strategy. Base token logits are computed by averaging predictions across all $k$ streams. Each interchangeable token's logit comes directly from its corresponding stream's projection. See Algorithm 5.

**Remark on batching.**   Reintroducing a batch dimension only requires reinstating a per-sequence mask that disables unused interchangeable-token pipelines. All other steps lift naturally over the batch axis.

**Edge case: no interchangeable tokens.**   When the input contains no interchangeable tokens (i.e., $k = 0$), the architecture creates a single stream ($k = 1$) that does not correspond to any specific interchangeable token. In this mode, the aggregated attention mechanisms operate identically to per-stream attention since there is only one stream to aggregate. Creating more than one stream in this case would be redundant: all operations are either per-stream (producing identical results across streams) or averaged across streams (where all streams would yield the same values due to the absence of interchangeable tokens). A consequence of this design is that if the input contains no interchangeable tokens, the model cannot generate interchangeable tokens in its output. However, this limitation does not affect the considered applications.

# B. Proof of Theorem 4.1

We prove that the proposed architecture is completely invariant to alpha-renaming. The key insight is that alpha-renaming merely permutes the ordering of the parallel streams, and all operations in the architecture either process streams independently or aggregate streams in a way that is invariant to permutation.

*Proof.* Let $f$ be an alpha-renaming function that fixes $\mathbb{V}_n$ and permutes $\mathbb{V}_i$. We show that for any input sequence $\boldsymbol{x}$, applying the model to the alpha-renamed input $\boldsymbol{x}' = f(\boldsymbol{x})$ yields an output $\hat{\boldsymbol{y}}' = M(\boldsymbol{x}')$ such that $f^{-1}(\hat{\boldsymbol{y}}') = \hat{\boldsymbol{y}}$, where $\hat{\boldsymbol{y}} = M(\boldsymbol{x})$.

---

**Algorithm 1** Embedding with Interchangeable Tokens

---

**Input:** token indices $\mathbf{x} \in \mathbb{N}^L$, embedding matrix $\mathbf{W} \in \mathbb{R}^{(V_n+2) \times d}$
**Output:** embeddings $\mathbf{E} \in \mathbb{R}^{k \times L \times d}$, mask $\mathbf{M} \in \{0,1\}^{k \times L}$
**Parameters:** $V_n$ = base vocabulary size, $k$ = number of interchangeable tokens

**for** $i = 0$ **to** $k - 1$ **do**
   $\mathbf{x}' \leftarrow \mathbf{x}$ {Copy input tokens}
   **for** each position $j$ in $\mathbf{x}'$ **do**
     **if** $\mathbf{x}'[j] \geq V_n$ **then**
       **if** $\mathbf{x}'[j] = V_n + i$ **then**
         $\mathbf{x}'[j] \leftarrow V_n$
       **else**
         $\mathbf{x}'[j] \leftarrow V_n + 1$
       **end if**
     **end if**
   **end for**
   $\mathbf{E}[i, :, :] \leftarrow \text{Lookup}(\mathbf{x}', \mathbf{W})$ {Embed modified tokens}
   $\mathbf{M}[i, :] \leftarrow (\mathbf{x} = V_n + i)$ {Create mask for $i$-th token}
**end for**
Return $\mathbf{E}, \mathbf{M}$

---

**Algorithm 2** Aggregating Interchangeable Token Streams

---

**Input:** hidden $\mathbf{H} \in \mathbb{R}^{k \times L \times d}$, token masks $\mathbf{A} \in \{0,1\}^{k \times L}$
**Output:** aggregated stream $\mathbf{G} \in \mathbb{R}^{L \times d}$
{Mean over parallel streams}
$\mathbf{g} \leftarrow \sum_i \mathbf{H}[i, :, :]$
$\mathbf{G} \leftarrow \mathbf{g}/k$
{Restore actual embeddings at their positions}
**for** $i = 0$ **to** $k - 1$ **do**
   $\mathbf{P} \leftarrow \mathbf{A}[i, :]$ {Positions of token $i$}
   $\mathbf{G} \leftarrow \mathbf{G} \cdot (1 - \mathbf{P}[:, None]) + \mathbf{H}[i, :, :] \cdot \mathbf{P}[:, None]$
**end for**
Return $\mathbf{G}$

---

**Algorithm 3** Encoder Layer with Parallel Interchangeable Streams

---

**Input:** hidden $\mathbf{H} \in \mathbb{R}^{k \times L \times d}$, self-attention mask $\mathbf{m}$, token masks $\mathbf{A}$
**Output:** updated hidden $\mathbf{H}^{out}$, self-attention weights
{Per-stream self-attention}
$\mathbf{S}, w_{self} \leftarrow \text{MHA}(\mathbf{H}, \mathbf{H}, \mathbf{H}, \mathbf{m})$
$\mathbf{H}_1 \leftarrow \text{Norm}(\mathbf{H} + \text{Dropout}(\mathbf{S}))$
{Attend to aggregated interchangeable tokens}
$\mathbf{G} \leftarrow \text{AggregateAPS}(\mathbf{H}_1, \mathbf{A})$
$\mathbf{P}, w_{ap} \leftarrow \text{MHA}(\mathbf{H}_1, \mathbf{G}, \mathbf{G}, \mathbf{m})$
$\mathbf{H}_2 \leftarrow \text{Norm}(\mathbf{H}_1 + \text{Dropout}(\mathbf{P}))$
{Position-wise feed-forward}
$\mathbf{F} \leftarrow \text{FFN}(\mathbf{H}_2)$
$\mathbf{H}^{out} \leftarrow \text{Norm}(\mathbf{H}_2 + \text{Dropout}(\mathbf{F}))$
Return $\mathbf{H}^{out}, w_{self}$

---

---

**Algorithm 4** Decoder Layer with Parallel Interchangeable Streams

---

**Input:** hidden $\mathbf{H} \in \mathbb{R}^{k \times L \times d}$, look-ahead mask $\mathbf{m}_{la}$, token masks $\mathbf{A}$, encoder hidden $\mathbf{E} \in \mathbb{R}^{k \times L_e \times d}$, encoder token masks $\mathbf{A}^e$, padding mask $\mathbf{m}_{pad}$, cross-attention modes list modes (entries `per` or `agg`)
**Output:** updated hidden $\mathbf{H}^{out}$, self-attention weights
{Masked per-stream self-attention}
$\mathbf{S}, w_{self} \leftarrow \text{MHA}(\mathbf{H}, \mathbf{H}, \mathbf{H}, \mathbf{m}_{la})$
$\mathbf{H}_1 \leftarrow \text{Norm}(\mathbf{H} + \text{Dropout}(\mathbf{S}))$
{Attend to aggregated interchangeable tokens}
$\mathbf{G} \leftarrow \text{AggregateAPS}(\mathbf{H}_1, \mathbf{A})$
$\mathbf{P}, w_{ap} \leftarrow \text{MHA}(\mathbf{H}_1, \mathbf{G}, \mathbf{G}, \mathbf{m}_{la})$
$\mathbf{H}_2 \leftarrow \text{Norm}(\mathbf{H}_1 + \text{Dropout}(\mathbf{P}))$
{Cross-attention stack (optional)}
$\mathbf{H}_x \leftarrow \mathbf{H}_2$
**for** mode in modes **do**
    **if** mode == `agg` **then**
        $\mathbf{K} \leftarrow \text{AggregateAPS}(\mathbf{E}, \mathbf{A}^e)$ {Aggregate encoder streams}
    **else**
        $\mathbf{K} \leftarrow \mathbf{E}$ {Per-stream encoder keys/values}
    **end if**
    $\mathbf{C} \leftarrow \text{MHA}(\mathbf{H}_x, \mathbf{K}, \mathbf{K}, \mathbf{m}_{pad})$
    $\mathbf{H}_x \leftarrow \text{Norm}(\mathbf{H}_x + \text{Dropout}(\mathbf{C}))$
**end for**
{Position-wise feed-forward}
$\mathbf{F} \leftarrow \text{FFN}(\mathbf{H}_x)$
$\mathbf{H}^{out} \leftarrow \text{Norm}(\mathbf{H}_x + \text{Dropout}(\mathbf{F}))$
Return $\mathbf{H}^{out}, w_{self}$

---

---

**Algorithm 5** Projection with Interchangeable Tokens

---

**Input:** hidden states $\mathbf{H} \in \mathbb{R}^{k \times L \times d}$, weight matrix $\mathbf{W} \in \mathbb{R}^{(V_n+2) \times d}$
**Output:** logits $\mathbf{Y} \in \mathbb{R}^{L \times (V_n+k)}$
**Parameters:** $V_n$ = base vocabulary size, $k$ = number of interchangeable tokens

{Project each parallel hidden state}
**for** $i = 0$ to $k-1$ **do**
    $\mathbf{Z}[i, :, :] \leftarrow \mathbf{H}[i, :, :]\mathbf{W}^T$ {Linear projection}
**end for**

{Combine logits}
Initialize $\mathbf{Y} \in \mathbb{R}^{L \times (V_n+k)}$

{Base tokens: average across parallel representations}
**for** $t = 0$ to $V_n - 1$ **do**
    $\mathbf{Y}[:, t] \leftarrow \frac{1}{k} \sum_{i=0}^{k-1} \mathbf{Z}[i, :, t]$
**end for**

{Interchangeable tokens: use corresponding representation}
**for** $i = 0$ to $k-1$ **do**
    $\mathbf{Y}[:, V_n + i] \leftarrow \mathbf{Z}[i, :, V_n]$ {Logit from actual embedding}
**end for**

Return $\mathbf{Y}$

---

**Step 1: Stream Permutation from Alpha-Renaming.** Consider alpha-renaming $f$ that maps interchangeable token $i$ to position $f(i)$ in the vocabulary $\mathbb{V}'_i$. When we embed the renamed input $\boldsymbol{x}'$ using Algorithm 1, the embedding algorithm creates $k$ streams where stream $j$ corresponds to interchangeable token $j$ in the renamed vocabulary. The key observation is that stream $j$ in the renamed input corresponds to stream $f^{-1}(j)$ in the original input. Thus, applying alpha-renaming causes the streams to be reordered via a permutation $\pi = f|_{\mathbb{V}_i}$ restricted to interchangeable tokens.

**Step 2: Per-Stream Operations Are Invariant to Reordering.** In both encoder and decoder layers (Algorithm 3 and Algorithm 4), each stream undergoes identical per-stream operations: self-attention, layer normalization, and feed-forward networks. All these operations process each stream independently using shared parameters (identical weights across all $k$ streams). Reordering the streams does not change the computation performed on each stream; stream $i$ receives the same input and applies the same transformations regardless of the ordering of streams in the batch.

Formally, if $\mathbf{H} \in \mathbb{R}^{k \times L \times d}$ is the input hidden state matrix and PerStreamOps denotes any per-stream operation (e.g., self-attention or feed-forward), then

$$\text{PerStreamOps}(\pi(\mathbf{H})) = \pi(\text{PerStreamOps}(\mathbf{H})) \tag{3}$$

where $\pi(\mathbf{H})$ denotes permuting the stream dimension by permutation $\pi$.

**Step 3: Aggregated Operations Are Invariant Due to Commutativity.** The aggregation step in Algorithm 2 is the key operation enabling inter-stream communication. It first averages hidden states across all $k$ streams and then restores actual embeddings at their designated positions:

$$\mathbf{G} = \frac{1}{k} \sum_{i=0}^{k-1} \mathbf{H}[i, :, :] \tag{4}$$

$$\mathbf{G} \leftarrow \mathbf{G} \cdot (1 - \mathbf{A}[i, :][:, \text{None}]) + \mathbf{H}[i, :, :] \cdot \mathbf{A}[i, :][:, \text{None}] \tag{5}$$

The sum operation in the averaging step is commutative: permuting the summands does not change the result. Additionally, the restoration loop iterates through interchangeable token positions marked by mask $\mathbf{A}[i, :]$, which specifies *where* token $i$ appears. Under alpha-renaming, the mask structure is permuted along with the streams: if token $i$ originally appeared at certain positions, then in the renamed input, token $f(i)$ appears at those same positions. The restoration loop correctly restores stream $f(i)$ at positions marked by $\mathbf{A}[f(i), :]$. Thus, aggregation produces the same output regardless of stream ordering:

$$\text{Aggregate}(\pi(\mathbf{H}), \pi(\mathbf{A})) = \text{Aggregate}(\mathbf{H}, \mathbf{A}) \tag{6}$$

**Step 4: Projection Preserves Invariance.** In Algorithm 5, base token logits are computed by averaging across all $k$ streams, which is commutative and unaffected by reordering. Interchangeable token $i$'s logit comes from stream $i$. Under alpha-renaming $f$, interchangeable token $f(i)$ appears at the same input positions as token $i$ originally, and logit for $f(i)$ is computed from stream $f(i)$, which corresponds to stream $i$ in the original computation. When $f^{-1}$ is applied to the output logits, interchangeable token $f(i)$ is mapped back to token $i$, correctly inverting the alpha-renaming.

**Conclusion.** Since each component of the architecture is invariant to stream reordering, and the renaming of streams is the only effect of alpha-renaming on the computation, we conclude that $M(\boldsymbol{x}') = f(M(\boldsymbol{x}))$, equivalently $f^{-1}(M(\boldsymbol{x}')) = M(\boldsymbol{x}) = \hat{\boldsymbol{y}}$. This completes the proof. $\qquad \square$

# C. Background: Formal Logic

This section provides an overview of the two formal logic systems studied in this work: propositional logic and Linear Temporal Logic (LTL). We begin with propositional logic and then present LTL as a temporal extension.

## C.1. Propositional Logic

Propositional logic provides a formal framework for reasoning about Boolean statements without considering time or sequence. Given a finite set $P$ of atomic propositions, where each $p \in P$ represents a Boolean variable, the syntax of

propositional formulae is defined as shown in Equation 7. Here, $\mathbf{T}$ denotes the constant *True*, and the operators include negation ($\neg$), conjunction ($\wedge$), disjunction ($\vee$), equivalence ($\leftrightarrow$), and exclusive or ($\oplus$).

$$\phi := \mathbf{T} \mid p \mid \neg\phi \mid \phi_1 \wedge \phi_2 \mid \phi_1 \vee \phi_2 \mid \phi_1 \leftrightarrow \phi_2 \mid \phi_1 \oplus \phi_2 \tag{7}$$

The satisfiability problem in propositional logic asks whether there exists a Boolean assignment to the atomic propositions in $P$ that makes a given formula $\phi$ evaluate to true. Our approach permits *partial* assignments, where only a subset of propositions receives explicit values while others remain unspecified. For instance, setting $a = 1$ alone suffices to satisfy $a \vee b$, leaving $b$ unconstrained.

We encode such assignments as alternating sequences of proposition symbols and their corresponding Boolean values. As an illustration, the string `a1b0` encodes the assignment where $a$ is set to true and $b$ is set to false. To validate model outputs and construct training datasets, we employ the `pyaiger` library (Vazquez-Chanlatte & Rabe).

### C.2. Linear Temporal Logic

Linear Temporal Logic (LTL) builds upon propositional logic by adding operators that express properties over time (Pnueli, 1977). The syntax of LTL formulae, defined over atomic propositions $P$, is given in Equation 8. In addition to the constant $\mathbf{T}$, atomic propositions $p \in P$, negation $\neg$, and conjunction $\wedge$ from propositional logic, LTL introduces two temporal operators: $\mathbf{X}$ (*next*) and $\mathbf{U}$ (*until*).

$$\phi := \mathbf{T} \mid p \mid \neg\phi \mid \phi_1 \wedge \phi_2 \mid \mathbf{X}\phi \mid \phi_1 \mathbf{U} \phi_2 \tag{8}$$

Note that propositional logic (Equation 7) includes derived operators such as equivalence and exclusive or, which are not present in LTL (Equation 8). This means that even though LTL builds upon propositional logic, the two systems pose different reasoning challenges, and one is not necessarily harder than the other.

The temporal operators have the following semantics:

- $\mathbf{X}\phi$: The formula $\phi$ must be satisfied at the immediately following time step (i.e., at time $t + 1$ when evaluated at time $t$).

- $\phi_1 \mathbf{U} \phi_2$: The formula $\phi_2$ must eventually hold at some future time step $t_2$, and $\phi_1$ must remain true at all intermediate time steps from the current time $t_1$ up to (but not necessarily including) $t_2$.

For example, $\mathbf{X}\mathbf{X}a$ specifies that proposition $a$ must be true at the third time step. The formula $\mathbf{T}\mathbf{U}a$ demands that $a$ becomes true at some point in the future. A more complex example is $\mathbf{X}b \wedge a\mathbf{U}c$, which requires $b$ to hold at the second step, $c$ to eventually hold, and $a$ to remain true at all time steps prior to when $c$ holds.

**Symbolic Traces.** A key distinction between LTL and propositional logic is that LTL formulae are evaluated over *symbolic traces*—sequences that specify truth values of atomic propositions across time steps. Following the approach in DeepLTL (Hahn et al., 2021), we consider symbolic traces of *infinite* length represented in *lasso* form, denoted $uv^\omega$. Here, $u$ is a finite prefix sequence, and $v$ is a finite sequence that repeats indefinitely to form the infinite suffix.

A symbolic trace captures *all* concrete traces that satisfy the propositional constraints at each respective time step. For example, the symbolic trace $a, a \wedge \neg b, (c)^\omega$ characterizes any trace where $a$ is true at the first two time steps, $b$ is false at the second time step, and $c$ is true from the third time step onward. This trace satisfies both $\mathbf{T}\mathbf{U}c$ and $\mathbf{X}\neg b \wedge a\mathbf{U}c$, but violates $\mathbf{X}\mathbf{X}b$ since $b$ is not required to be true at the third time step.

Symbolic traces may be *underspecified*, meaning certain propositions can assume arbitrary values at certain time steps. In the example above, the value of $a$ at time steps beyond the second is not constrained, and similarly for $b$ beyond the second step.

**Problem Formulation and Notation.** The LTL solving problem is to find a symbolic trace in lasso form $uv^\omega$ that satisfies a given formula $\phi$. We formulate this as an autoregressive generation task: conditioned on an LTL formula and a partially generated symbolic trace, the model predicts the distribution over possible next tokens.

For consistency with the DeepLTL dataset (Hahn et al., 2021), we represent both formulae and traces using Polish (prefix) notation, where operators precede their operands. For instance, $a \land b$ is written as `&ab`, eliminating the need for parentheses.

Traces in our notation include atomic propositions, the constants `True:1` and `False:0`, logical operators, and special delimiters: ";" separates time steps, while "{" and "}" enclose the repeating suffix $v$. As an example, the string "`a;&ab;{b}`" encodes the symbolic trace $a, a \land b, (b)^\omega$.

## D. Hyperparameters

Table 5 lists the hyperparameter choices for all experiments and models. The hyperparameters for the baselines are taken from the prior work (Hahn et al., 2021; Işık et al., 2025). As seen in the table, we used smaller embedding sizes for our proposed method compared to the baselines based on preliminary hyperparameter tuning that showed competitive performance with reduced model capacity. This highlights superior parameter efficiency of our method compared to the baselines. We increased the feed-forward layer size of the proposed method on the propositional logic task, but we did not observe any improvement when we did the same for the baselines. For example, 10 AP validation accuracy of the full vocabulary baseline on the propositional logic task was 87.87% with 512 FC size and 87.36% with 768 FC size.

*Table 5.* Hyperparameter choices. Differing values between the baselines and our proposed method are highlighted in bold.

| Model | Experiment | Embedding | Layers | Heads | FC size | Batch Size | Train Steps |
|---|---|---|---|---|---|---|---|
| Baselines | Copying | 128 | 6 | 8 | 128 | 512 | 20K |
| | LTL | 128 | 8 | 8 | 1024 | 768 | 52K |
| | Propositional Logic | 132 | 6 | 6 | 512 | 1024 | 50K |
| Proposed Method | Copying | 128 | 6 | 8 | 128 | 512 | 20K |
| | LTL | **64** | 8 | **4** | 1024 | 768 | 52K |
| | Propositional Logic | **96** | 6 | 6 | **768** | 1024 | 50K |

## E. Copying with Scalable Vocabulary

We use the training dataset from the prior work (Işık et al., 2025) comprising 10 million random strings with lengths ranging from 20 to 80 characters and vocabulary sizes up to 20 unique characters. The test dataset contains 20 samples for each feasible combination of sequence length and unique character count, arranged in an upper-triangular matrix structure (since unique character count cannot exceed sequence length). We focus exclusively on this larger-scale variant for several reasons. First, experiments on smaller copying tasks (with shorter sequences and smaller vocabularies) exhibited high variance and yielded inconclusive results in prior work. Second, unlike the random embedding approach which required extensive hyperparameter exploration across embedding methods and normalization strategies, the proposed method introduces minimal hyperparameters, obviating the need for toy-scale parameter searches. Third, demonstrating robust performance on a more challenging generalization scenario provides stronger evidence of the method's capabilities.

Unlike the random embedding baseline which requires evaluation across multiple random seeds to characterize variability, the proposed method is fully deterministic (apart from floating-point numerical precision) and produces identical predictions given the same input. This eliminates sensitivity to embedding randomization entirely.

The evaluation framework employs edit distance to measure alignment between model predictions and ground truth sequences. On this task, the proposed method achieves perfect performance with mean edit distance of 0.0, matching the performance of all baseline configurations. In the next subsections, we focus on more complex and challenging logic-based tasks.

## F. Addressing Imbalance in LTL Dataset

For the LTL task, we identify an imbalance in the training set that affects the performance in 0 AP and 1 AP scenarios. Specifically, the training dataset is underrepresented in these cases, which affects performance of the baselines. We augment the training data to mitigate this issue and re-train all LTL models in Section 5.4 for a fair comparison.

To ensure no overlap between the training and evaluation sets, we take the 0 AP and 1 AP samples from the full vocabulary

baseline's training set and add them to the original training set. This increases the 0 AP samples from 1 to 203 and the 1 AP samples from 11,635 to 22,474. The distribution change is illustrated in Figure 3. The heatmap evaluation of the alpha-renaming baseline before and after augmentation is shown in Figure 4. As seen in the figure, the model fails on almost all 0 AP cases and struggles with 1 AP cases before augmentation. But after addressing the imbalance, the performance matches the rest of the region within the AP count & length boundary.

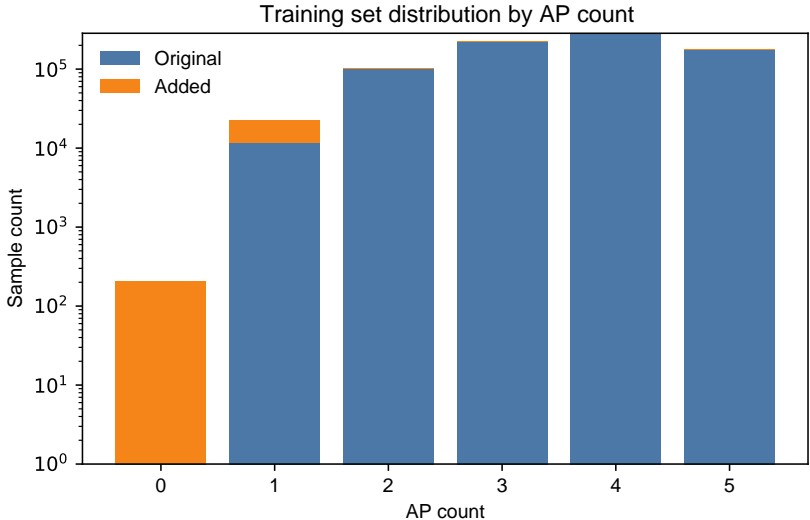

*Figure 3.* LTL training set distribution before and after augmentation in log scale.

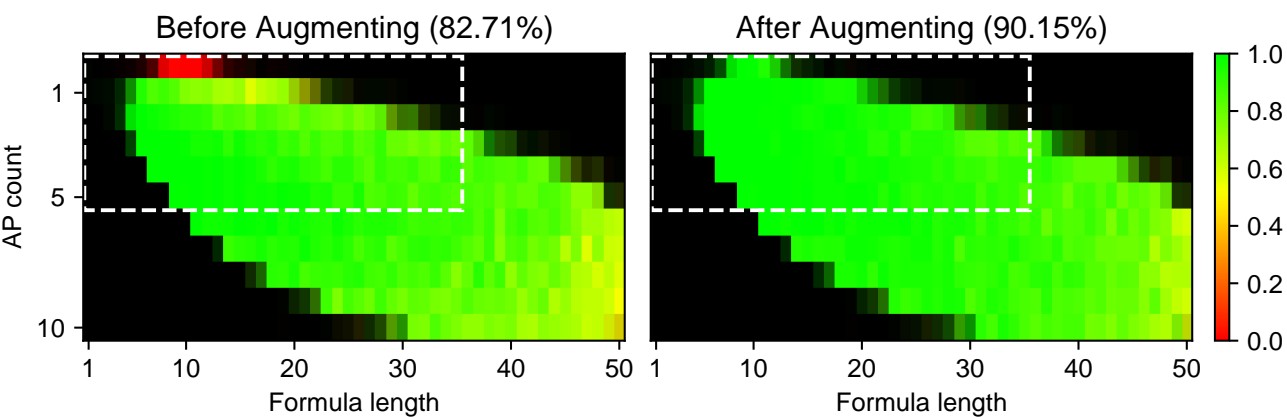

*Figure 4.* Heatmap evaluation of the alpha-renaming baseline before and after augmenting the training set to address imbalance.

## G. Detailed Ablation Studies

In this appendix, we provide the complete set of ablation heatmaps for both LTL solving and propositional logic assignment prediction tasks. Figure 6 shows the ablation heatmaps for LTL solving, while Figure 5 presents the ablation heatmaps for propositional logic assignment prediction. Each heatmap visualizes model accuracy across two dimensions: formula length (x-axis, 1–50) and argument pack count (y-axis). The left panel displays a table indicating which attention components are enabled for each configuration, including Encoder/Decoder/Cross-attention in both Per-Stream and Aggregated variants, along with model parameters and rank. The main heatmap uses a red-green colormap to show accuracy values (0 to 1), with black regions indicating areas lacking training data. The title reports overall accuracy across the evaluated configurations.

The abbreviations used in the configuration labels indicate which attention components are enabled:

- **EP**: Encoder Per-Stream attention

- **EA**: Encoder Aggregated attention

- **DP**: Decoder Per-Stream attention

- **DA**: Decoder Aggregated attention

- **CP**: Cross Per-Stream attention

- **CA**: Cross Aggregated attention

Table 6 summarizes the full ablation results, listing each configuration along with its corresponding validation accuracies on 5 and 10 AP datasets, heatmap accuracy, model parameter count, and figure reference.

*Table 6.* Full ablation results for LTL solving and propositional logic assignment prediction tasks. Columns show: (1) task type, (2) enabled attention components, (3-4) validation accuracy on 5 and 10 AP datasets, (5) heatmap accuracy, (6) model parameter count, and (7) corresponding figure reference. The models are sorted by task and heatmap accuracy in descending order.

| Task | Components | | | | | | 5 AP Val | 10 AP Val | Heatmap | Parameters | Figure |
|---|---|---|---|---|---|---|---|---|---|---|---|
| Propositional Logic | EP | DP | EA | DA | CP |    | 97.94% | 97.63% | 95.05% | 2,906,496 | 5b |
|  | EP | DP | EA | DA | CP | CA | 96.75% | 96.05% | 92.66% | 3,131,136 | 5a |
|  | EP | DP |    | DA | CP |    | 96.84% | 96.19% | 92.47% | 2,681,856 | 5g |
|  |    | DP | EA | DA | CP |    | 96.29% | 95.16% | 91.44% | 2,681,856 | 5o |
|  | EP | DP |    | DA | CP | CA | 95.69% | 94.95% | 91.10% | 2,906,496 | 5f |
|  |    | DP | EA | DA | CP | CA | 94.81% | 93.48% | 89.15% | 2,906,496 | 5n |
|  | EP | DP | EA |    | CP |    | 96.81% | 85.77% | 84.48% | 2,681,856 | 5e |
|  | EP | DP | EA |    | CP | CA | 96.70% | 85.76% | 84.33% | 2,906,496 | 5d |
|  | EP | DP |    |    | CP | CA | 96.00% | 83.40% | 82.16% | 2,681,856 | 5h |
|  | EP | DP |    |    | CP |    | 91.47% | 69.97% | 72.35% | 2,457,216 | 5i |
|  |    | DP | EA |    | CP |    | 88.96% | 67.74% | 70.33% | 2,457,216 | 5q |
|  |    | DP | EA |    | CP | CA | 88.38% | 66.32% | 68.99% | 2,681,856 | 5p |
|  |    |    | EA | DA | CP |    | 64.52% | 52.11% | 52.66% | 2,457,216 | 5s |
|  | EP |    | EA | DA | CP | CA | 59.21% | 43.89% | 47.82% | 2,906,496 | 5j |
|  |    |    | EA | DA | CP | CA | 60.58% | 47.60% | 47.14% | 2,681,856 | 5r |
|  | EP |    | EA | DA | CP |    | 56.94% | 44.09% | 46.55% | 2,681,856 | 5k |
|  | EP |    |    | DA | CP |    | 54.50% | 44.35% | 44.50% | 2,457,216 | 5m |
|  | EP | DP | EA | DA |    | CA | 39.95% | 20.86% | 28.51% | 2,906,496 | 5c |
|  | EP |    |    | DA | CP | CA | 36.17% | 26.15% | 26.91% | 2,681,856 | 5l |
| LTL | EP | DP | EA |    | CP |    | 98.12% | 96.53% | 90.47% | 2,654,144 | 6e |
|  | EP | DP | EA |    | CP | CA | 98.23% | 96.49% | 90.27% | 2,788,288 | 6d |
|  | EP | DP |    | DA | CP |    | 97.96% | 96.32% | 89.66% | 2,654,144 | 6g |
|  | EP | DP | EA | DA | CP |    | 98.33% | 96.45% | 89.47% | 2,788,288 | 6b |
|  | EP | DP |    |    | CP | CA | 97.99% | 96.05% | 89.26% | 2,654,144 | 6h |
|  | EP | DP | EA | DA | CP | CA | 97.96% | 96.32% | 89.16% | 2,922,432 | 6a |
|  | EP | DP |    | DA | CP | CA | 98.15% | 96.00% | 88.95% | 2,788,288 | 6f |
|  |    | DP | EA | DA | CP |    | 97.54% | 95.48% | 87.53% | 2,654,144 | 6o |
|  |    | DP | EA | DA | CP | CA | 97.68% | 95.39% | 87.40% | 2,788,288 | 6n |
|  | EP | DP |    |    | CP |    | 94.77% | 91.34% | 84.13% | 2,520,000 | 6i |
|  |    | DP | EA |    | CP |    | 94.90% | 90.99% | 84.13% | 2,520,000 | 6q |
|  |    | DP | EA |    | CP | CA | 94.78% | 90.84% | 82.76% | 2,654,144 | 6p |
|  | EP |    | EA | DA | CP |    | 18.64% | 22.00% | 21.13% | 2,654,144 | 6k |
|  | EP | DP | EA | DA |    | CA | 25.78% | 20.52% | 20.93% | 2,788,288 | 6c |
|  |    |    | EA | DA | CP |    | 17.44% | 19.01% | 20.27% | 2,520,000 | 6s |
|  | EP |    |    | DA | CP | CA | 18.81% | 20.98% | 19.60% | 2,654,144 | 6l |
|  | EP |    |    | DA | CP |    | 15.45% | 20.01% | 17.58% | 2,520,000 | 6m |
|  | EP |    | EA | DA | CP | CA | 14.20% | 16.53% | 16.92% | 2,788,288 | 6j |
|  |    |    | EA | DA | CP | CA | 14.34% | 15.02% | 15.99% | 2,654,144 | 6r |

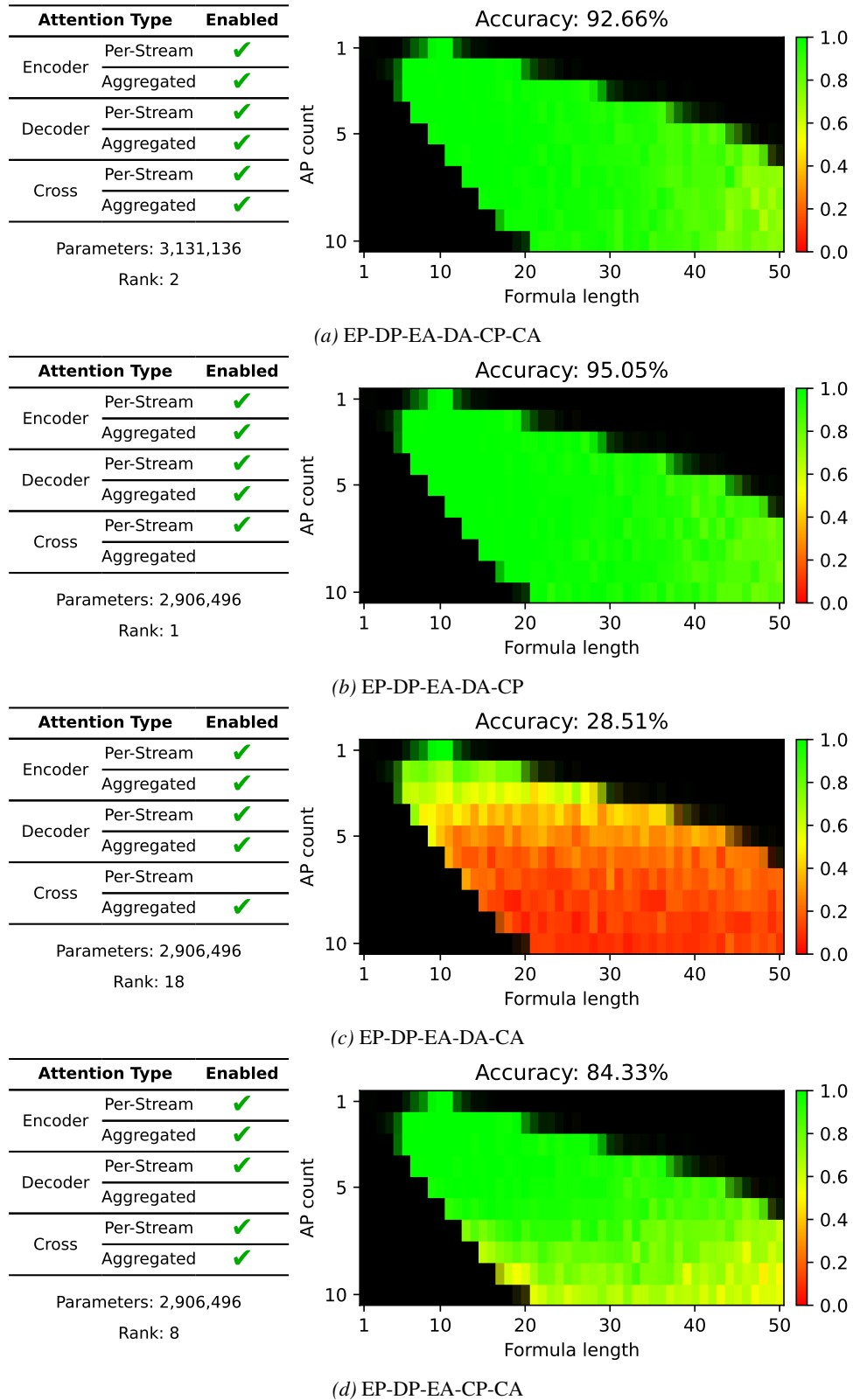

*Figure 5.* Ablation heatmaps for propositional logic assignment prediction.

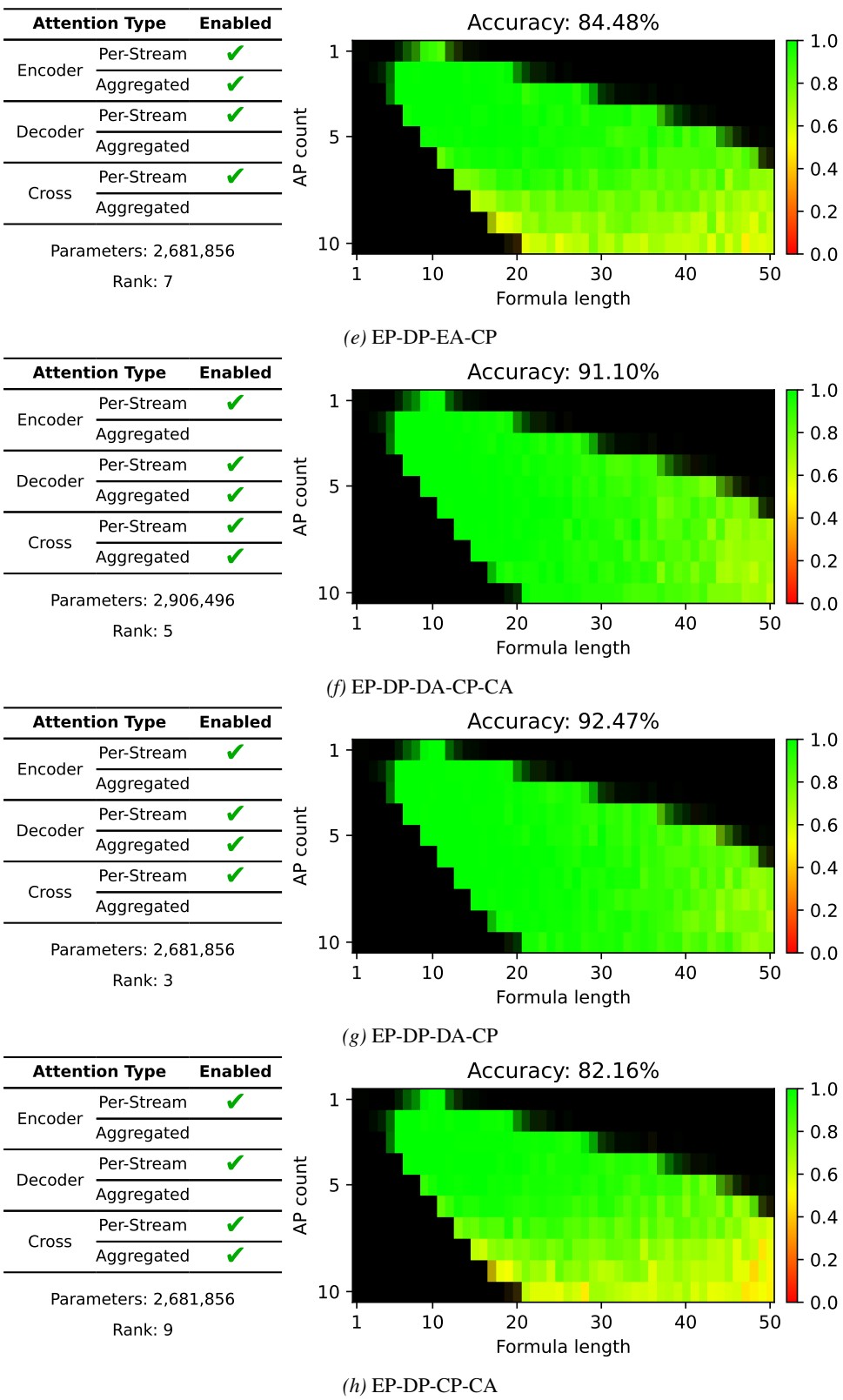

*Figure 5.* Ablation heatmaps for propositional logic assignment prediction.

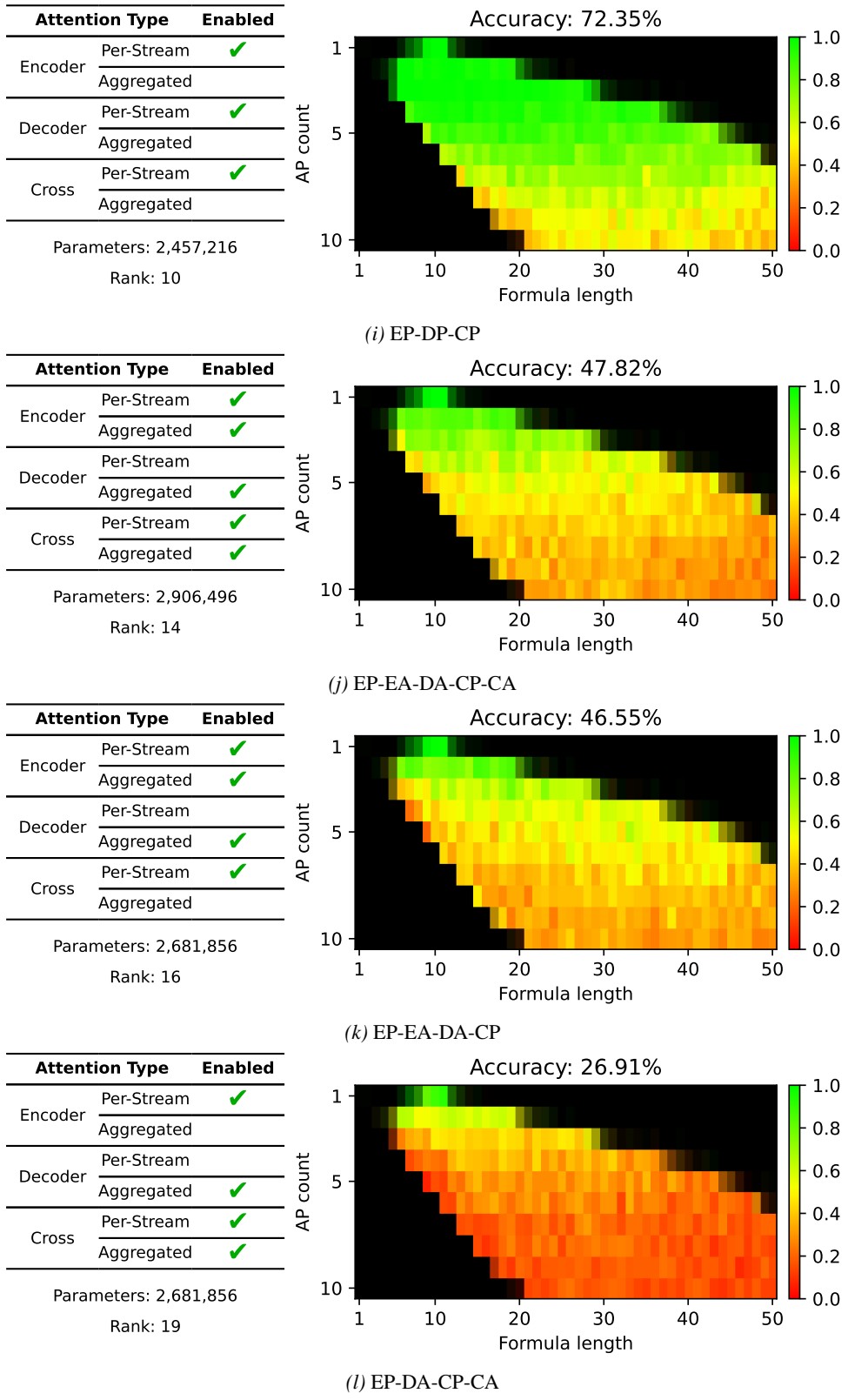

*Figure 5.* Ablation heatmaps for propositional logic assignment prediction.

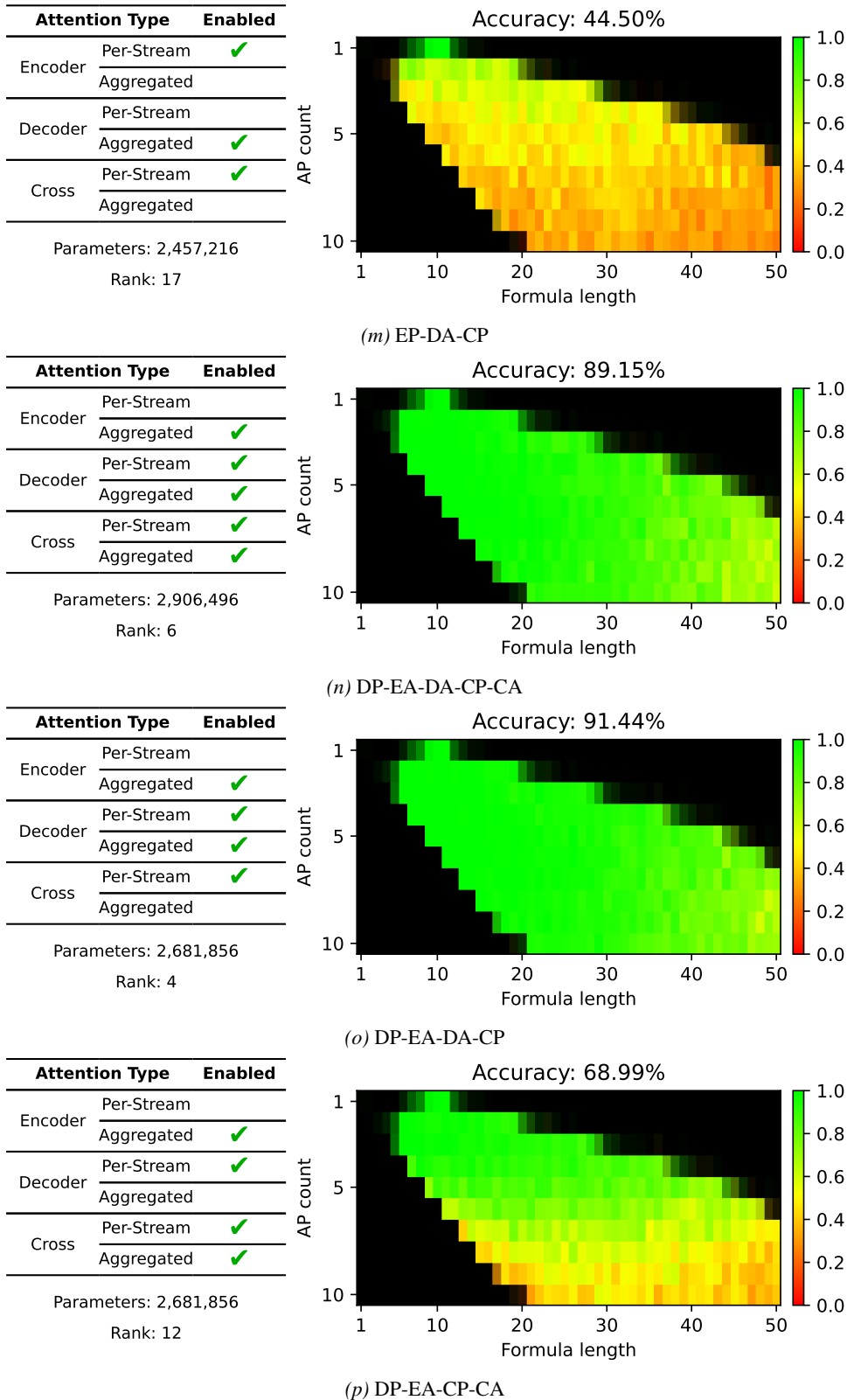

*Figure 5.* Ablation heatmaps for propositional logic assignment prediction.

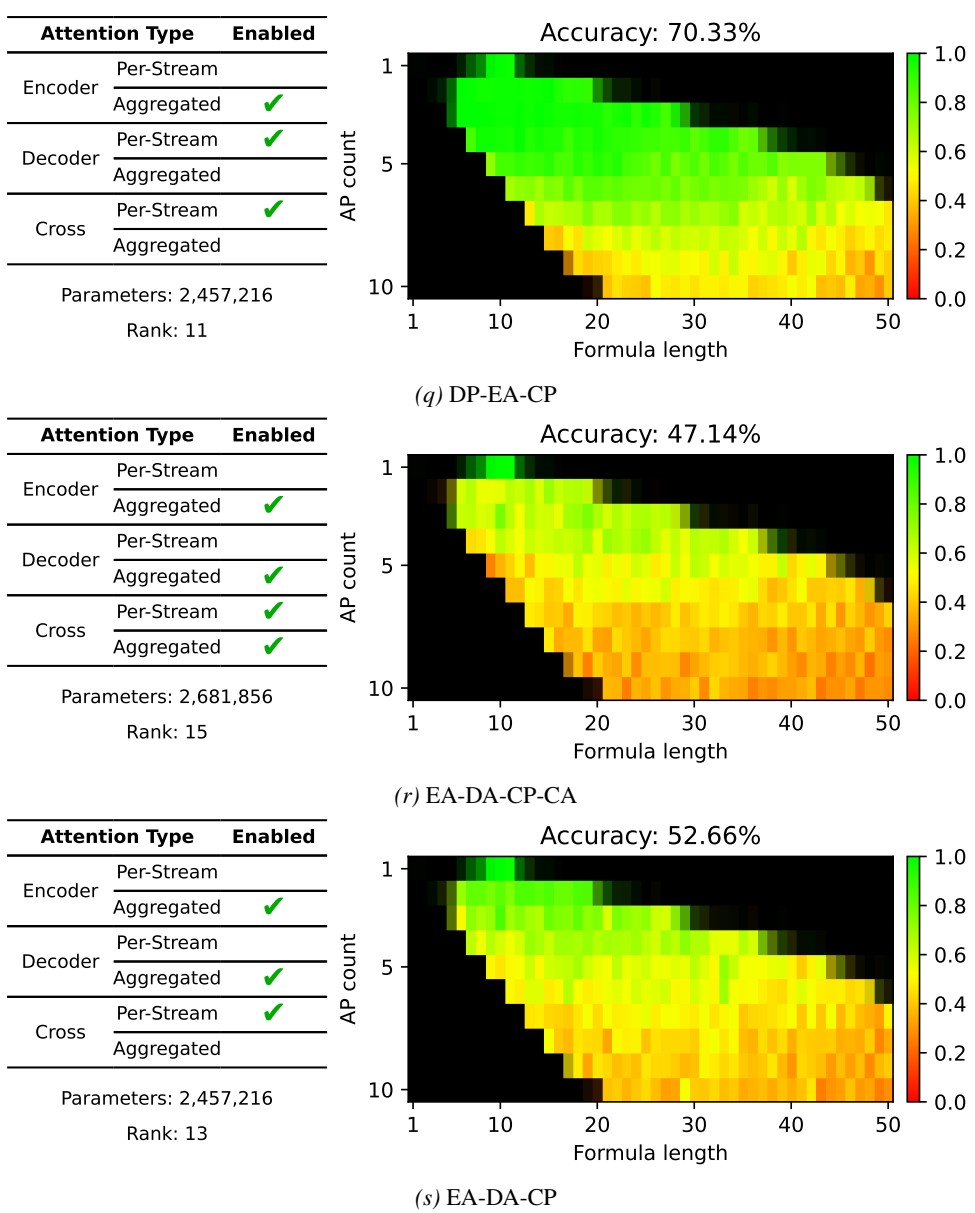

*Figure 5.* Ablation heatmaps for propositional logic assignment prediction.

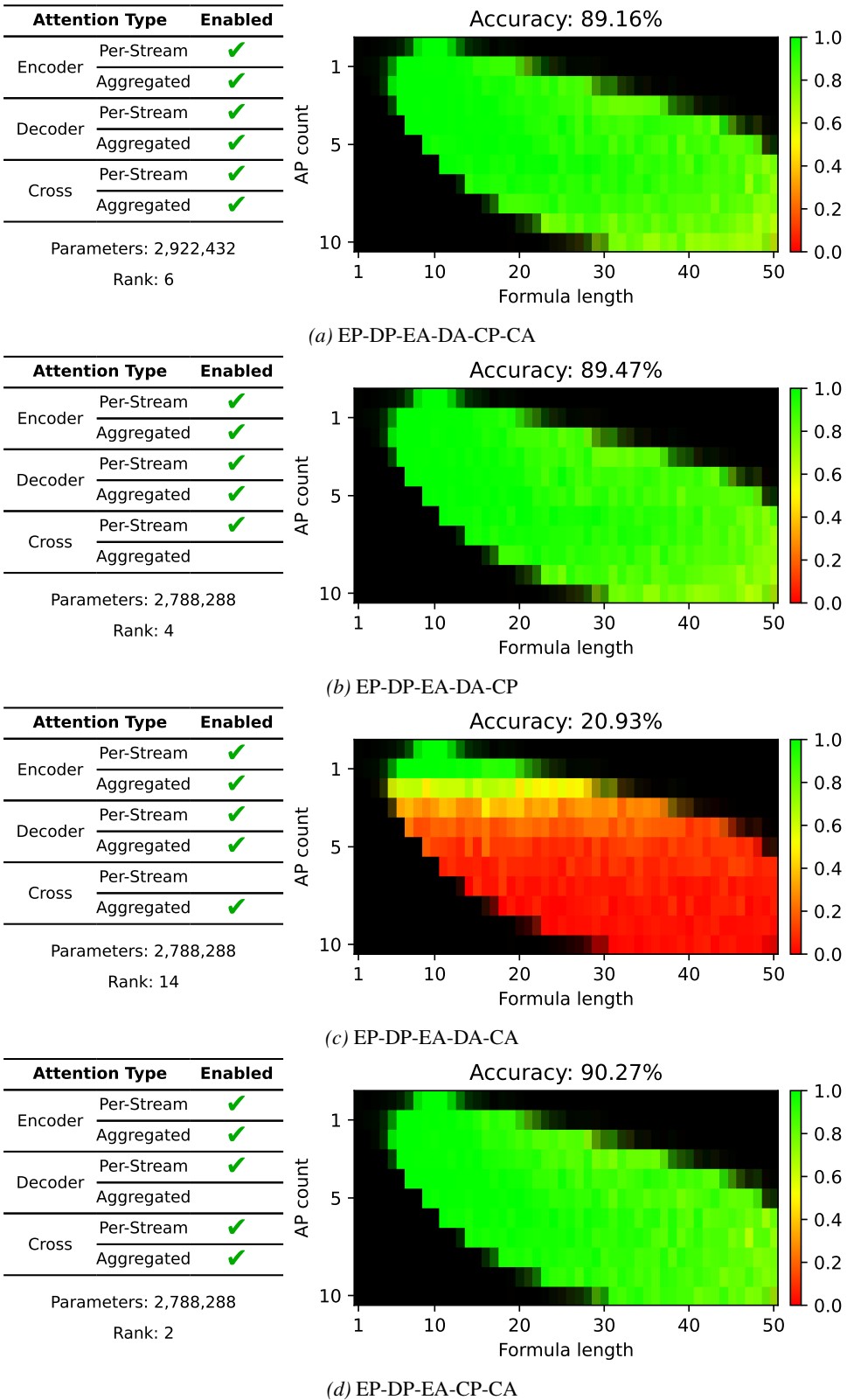

*Figure 6.* Ablation heatmaps for LTL solving.

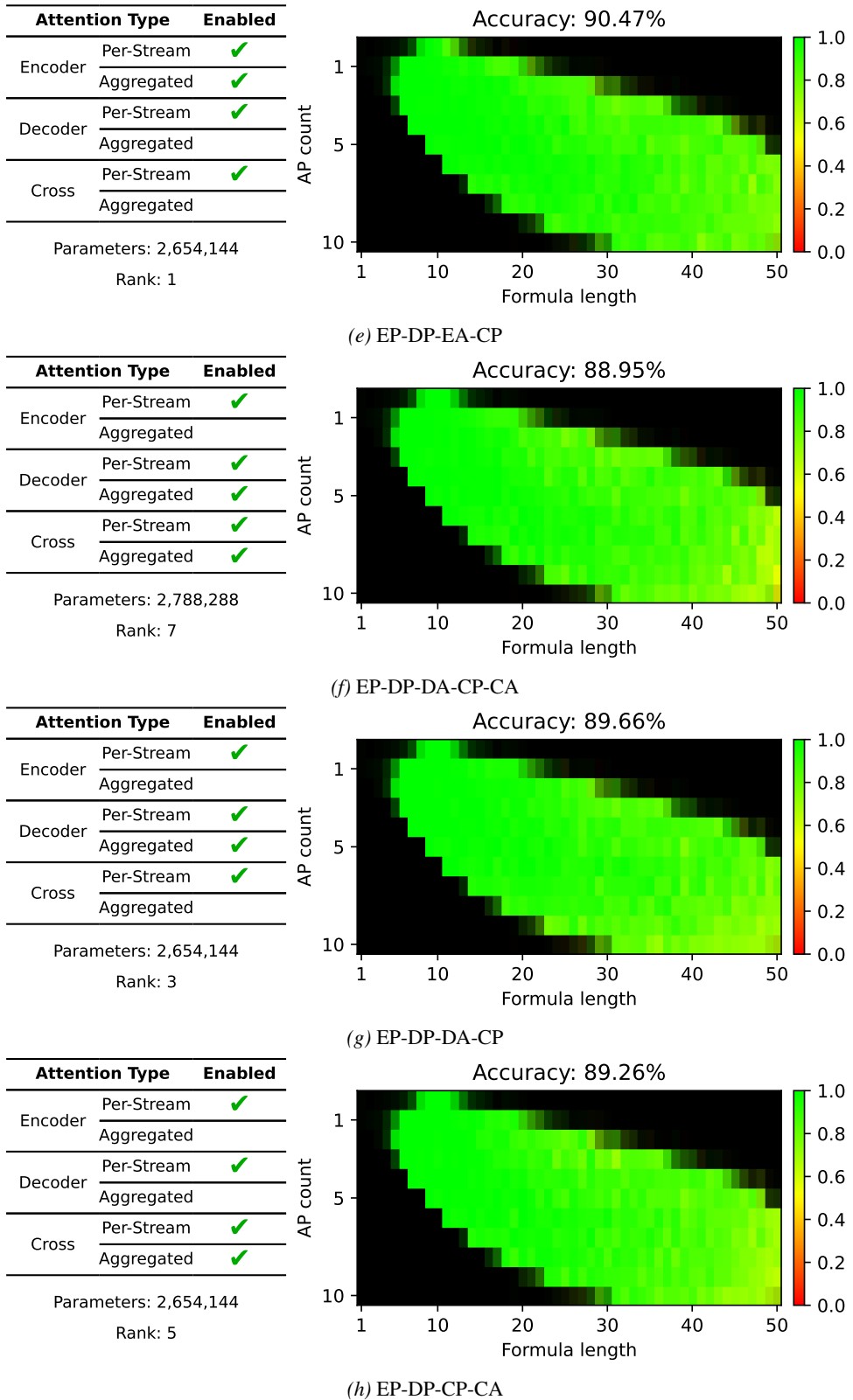

*Figure 6.* Ablation heatmaps for LTL solving.

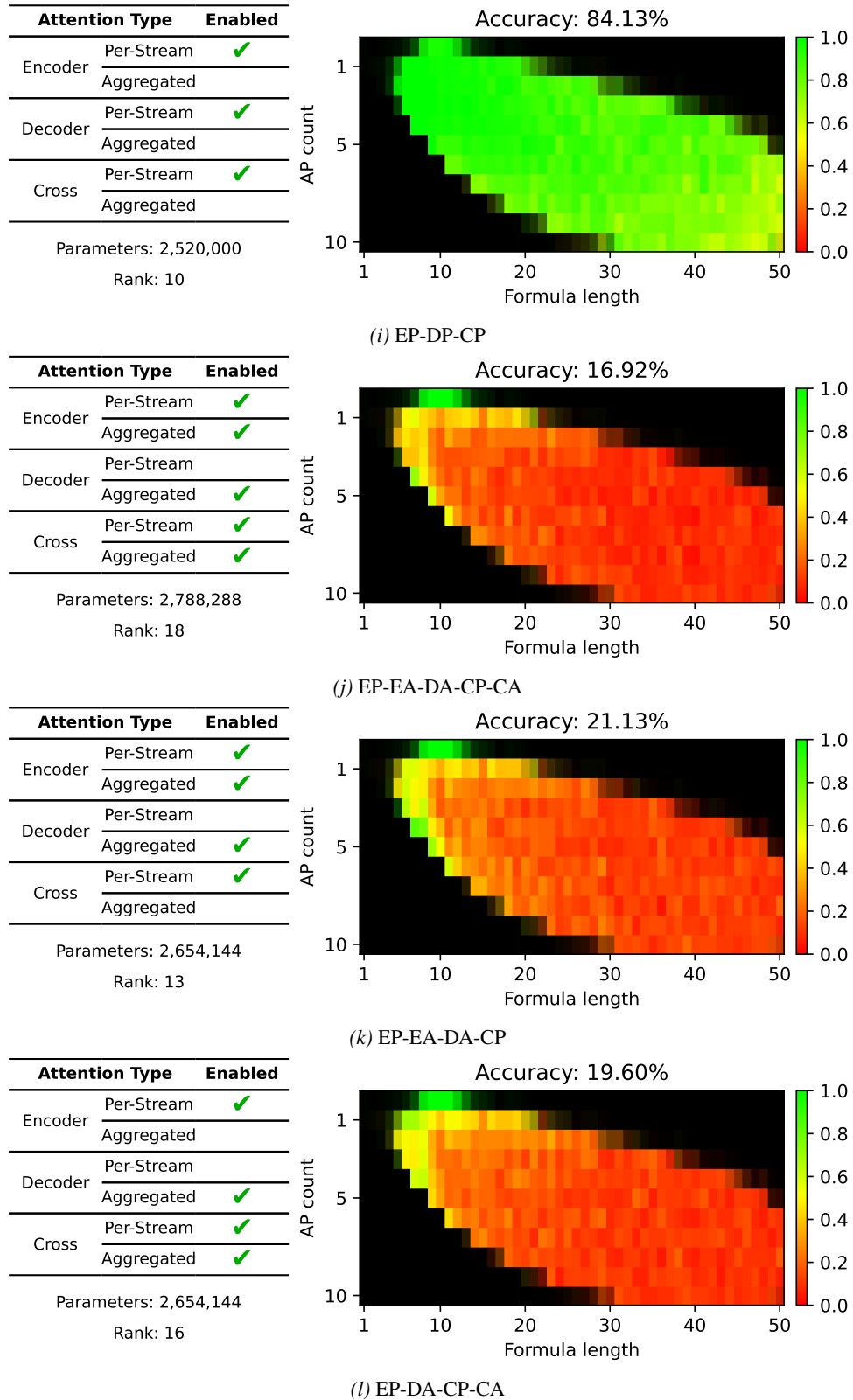

*Figure 6.* Ablation heatmaps for LTL solving.

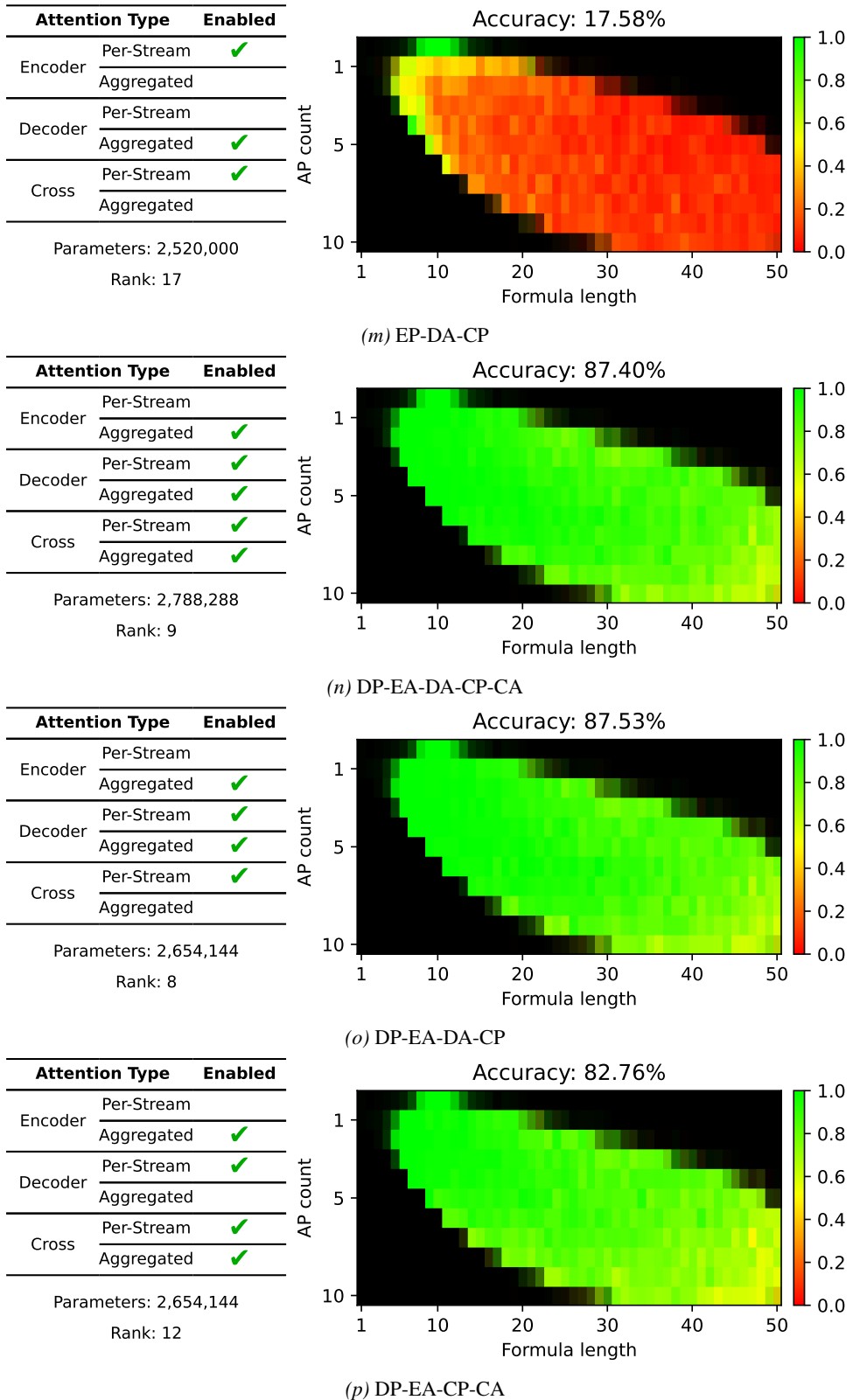

*(m)* EP-DA-CP

*(n)* DP-EA-DA-CP-CA

*(o)* DP-EA-DA-CP

*(p)* DP-EA-CP-CA

*Figure 6.* Ablation heatmaps for LTL solving.

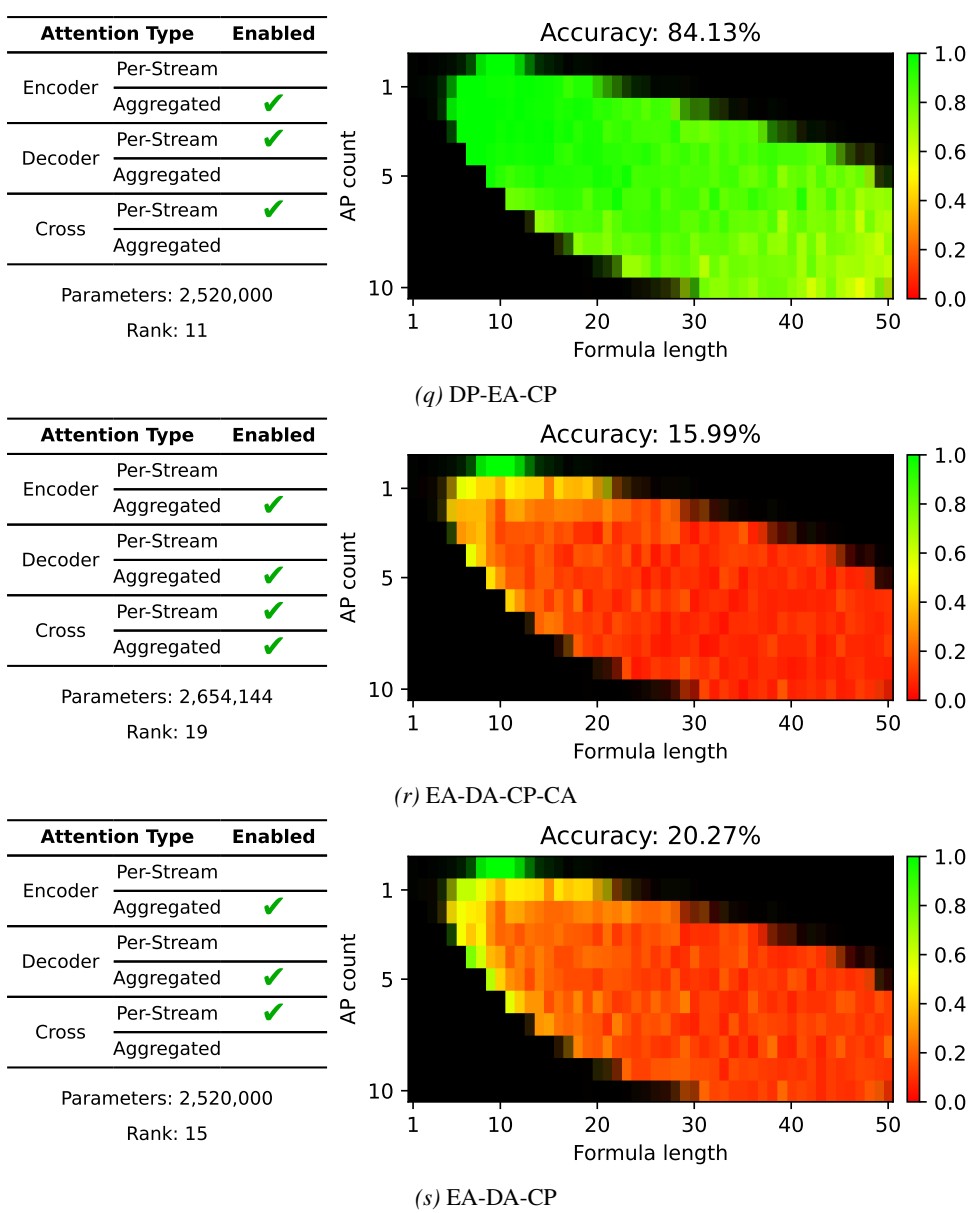

*Figure 6.* Ablation heatmaps for LTL solving.

# H. LLM Setup

For the experiments involving large language models (LLMs), we use GPT-5.2 (Singh et al., 2025) via the OpenAI Responses API (`v1/responses`).

While our specialized models process formulas in prefix (Polish) notation, we interact with the LLM using infix notation, which aligns better with natural language conventions and the expectations of general-purpose language models. For the propositional logic task, the LLM's responses are constrained to JSON format through schema validation. The specific prompts for both tasks are presented in Listings 1 and 2. During inference, the "`{formula}`" placeholder within each prompt is dynamically substituted with the concrete formula instance before being sent to the LLM.

We set the reasoning effort to medium and text verbosity to low in all experiments while using the default values for other parameters, except for the alpha-covariance evaluation. Since alpha-covariance does not evaluate the output correctness, we disable reasoning entirely to save costs, and we set the temperature to 0 for deterministic sampling.

With medium reasoning effort, GPT-5.2 takes 10 to 90 seconds to solve each sample in our benchmarks. In particular, GPT-5.2 takes 4 hours to solve 387 samples from the LTL heatmap test set (sequentially, without using the Batch API), averaging around 37 seconds per sample. We use the Batch API for the rest of our experiments, which doesn't report per-sample latency.

Due to financial constraints, we reduce the evaluation sample counts as follows:

- **Heatmap evaluations:** We evaluate on 10 samples instead of 100 per cell (AP count & length combination).

- **Test set in Table 1:** Sample size is reduced from 100k to 3k.

- **Alpha-covariance evaluations:** Sample size is reduced from 1k to 100.

The heatmaps for GPT-5.2 on the LTL and propositional logic tasks are shown in Figure 7.

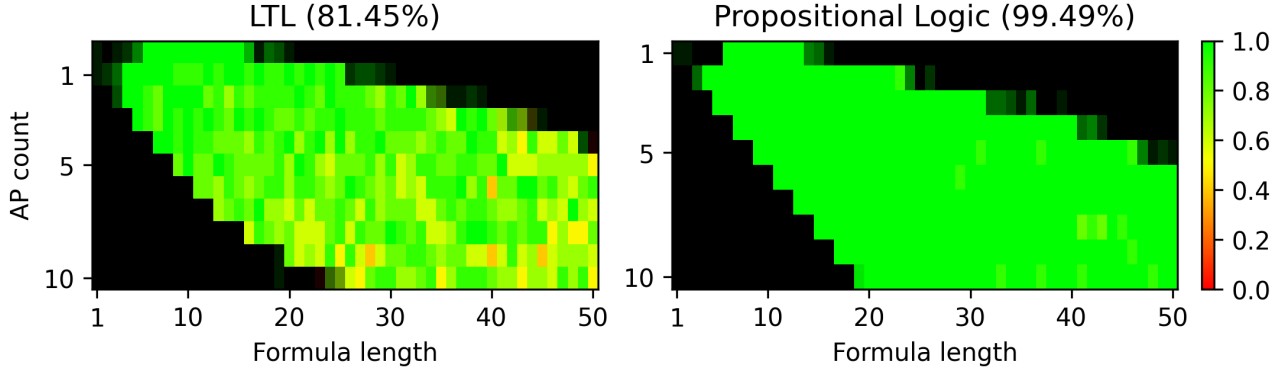

*Figure 7.* Heatmaps for GPT-5.2 on the LTL (left) and propositional logic (right) tasks.

*Listing 1.* LLM Prompt for the LTL solving task.

```
Your task is to generate a satisfying trace for a given LTL (Linear Temporal Logic)
    formula.
Lowercase letters denote the atomic propositions.
The output trace should be in lasso form composed of two parts: the prefix part and the
    cycle part.
Timesteps in the trace should be separated by semicolons, and the cycle part should be
    enclosed in curly braces, preceeded by the keyword "cycle".

Temporal operators:
X: Next operator
U: Until operator

Logical operators:
```

```
&: AND operator                                                                   11
|: OR operator                                                                    12
!: NOT operator                                                                   13
The output trace is a symbolic trace, which means that the logical operators are allowed,  14
    but not temporal operators.                                                   15
                                                                                  
Constants:                                                                        16
0: False                                                                          17
1: True                                                                           18
Note that other numbers are invalid.                                              19
                                                                                  20
Example 1                                                                         21
Formula: X((a & Xa) U XXb)                                                        22
Trace: 1; 1; 1; b; cycle{{1}}                                                     23
                                                                                  24
Example 2                                                                         25
Formula: !c U X(1 U b)                                                            26
Trace: 1; b; cycle{{1}}                                                           27
                                                                                  28
Example 3                                                                         29
Formula: X!X!(b & Xb)                                                             30
Trace: 1; 1; b; b; cycle{{1}}                                                     31
                                                                                  32
Example 4                                                                         33
Formula: !(1 U !c)                                                                34
Trace: cycle{{c}}                                                                 35
                                                                                  36
Your Turn                                                                         37
Formula: {formula}                                                                38
Please generate the corresponding trace. Output the trace only.                   39
```

*Listing 2.* LLM Prompt for the propositional logic task.

```
Your task is to generate an assignment that satisfies a given propositional logic formula.  1
Lowercase letters denote the atomic propositions.                                 2
The output is a JSON object representing the assignment.                          3
                                                                                  4
Logical operators (ordered from highest precedence to lowest):                    5
!: NOT operator                                                                   6
&: AND operator                                                                   7
|: OR operator                                                                    8
xor: Exclusive OR operator                                                        9
<->: Logical equivalence operator (biconditional)                                10
                                                                                  11
Constants:                                                                        12
0: False                                                                          13
1: True                                                                           14
Note that other numbers are invalid.                                              15
                                                                                  16
Example 1                                                                         17
Formula: !a | c & (b <-> c)                                                       18
Assignment: { "a": false }                                                        19
                                                                                  20
Example 2                                                                         21
Formula: !(a <-> (!a xor !e))                                                     22
Assignment: { "a": true, "e": true }                                             23
                                                                                  24
Example 3                                                                         25
Formula: a & (!a <-> !c | d)                                                      26
Assignment: { "a": true, "c": true, "d": false }                                 27
                                                                                  28
Example 4                                                                         29
Formula: !(a | !(!d | b & d))                                                     30
Assignment: { "a": false, "d": false }                                           31
                                                                                  32
```

```
Your Turn                                                                     33
Formula: {formula}                                                            34
Please generate an assignment that satisfies this formula. Output the assignment only, in    35
    JSON format.
```

# I. Top-N Accuracy

To compute the top-N accuracy, we generate multiple candidate solutions for each input formula by using beam search with a beam width of N. A candidate solution is considered correct if it satisfies the input formula. The top-N accuracy is then calculated as the percentage of input formulas for which at least one of the N candidate solutions is correct.

The top-N accuracy on the propositional logic heatmaps is shown in Figure 8 for the best proposed model.

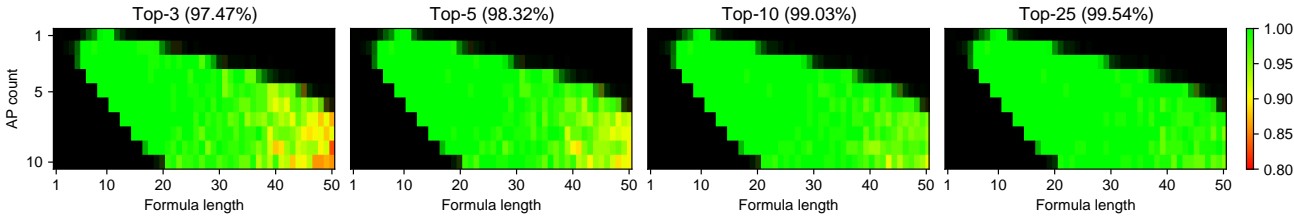

*Figure 8.* Top-N accuracy heatmaps for the best proposed model on the propositional logic task. Note that unlike other heatmaps, the color scale starts from 0.8 to better visualize differences at high accuracy levels.

# J. Computational Cost Analysis

In this appendix, we provide an analysis of the computational cost of our proposed method during inference. As discussed in Section 5.7, our architecture employs $S$ parallel streams to handle interchangeable tokens, leading to a time complexity of $O(SL^2)$ for a sequence of length $L$. This is in contrast to the standard attention mechanism's $O(L^2)$ complexity. To empirically validate this scaling behavior, we measure the average time required to generate a single sample during inference as a function of the number of interchangeable tokens (atomic propositions, APs) on the propositional logic task. Figure 9 presents the results of this profiling. As seen in the figure, the time scales linearly with the number of APs, consistent with the theoretical analysis.

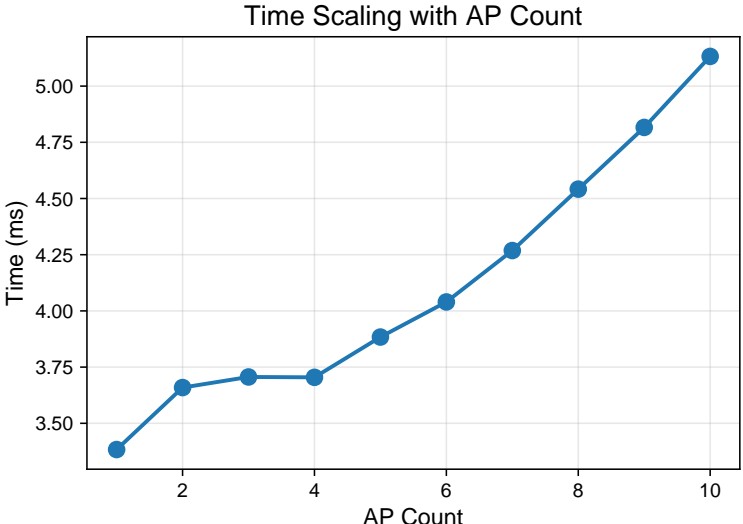

*Figure 9.* Average time required to generate a single sample during inference as a function of the number of interchangeable tokens (APs) for our proposed method. The measurements were taken on the propositional logic task using a single NVIDIA L40S GPU.

# K. Heatmaps for Converted & Fine-Tuned Models

This appendix presents the full out-of-distribution heatmaps for the fine-tuned models described in Section 5.8. Each heatmap visualizes model accuracy across formula length (x-axis, 1–50) and argument pack count (y-axis), using a red-green colormap.

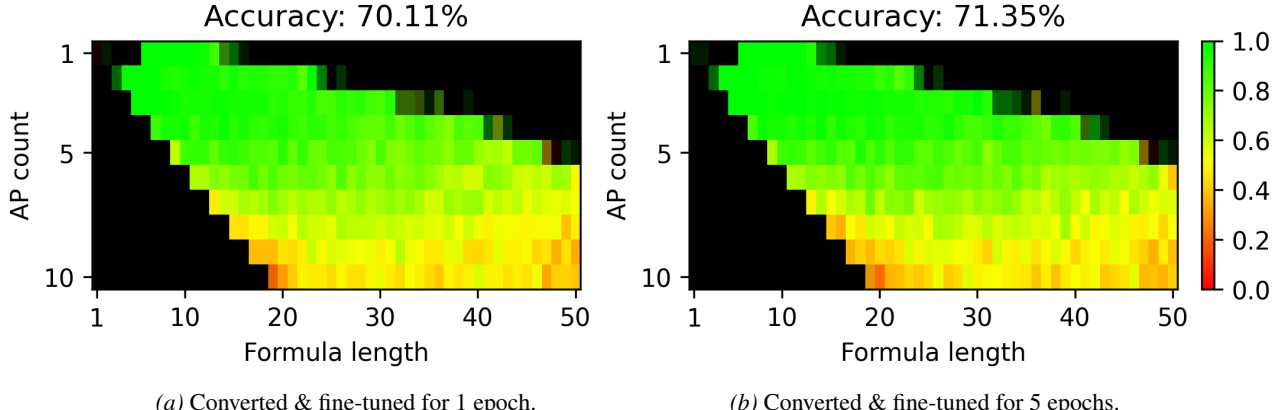

(a) Converted & fine-tuned for 1 epoch.
(b) Converted & fine-tuned for 5 epochs.

*Figure 10.* Heatmaps for the fine-tuned models on the propositional logic task. Both models are converted from the same pre-trained baseline. After one epoch of fine-tuning (a), the model approaches the from-scratch baseline (see Table 4). Five epochs of fine-tuning (b) further close the gap.

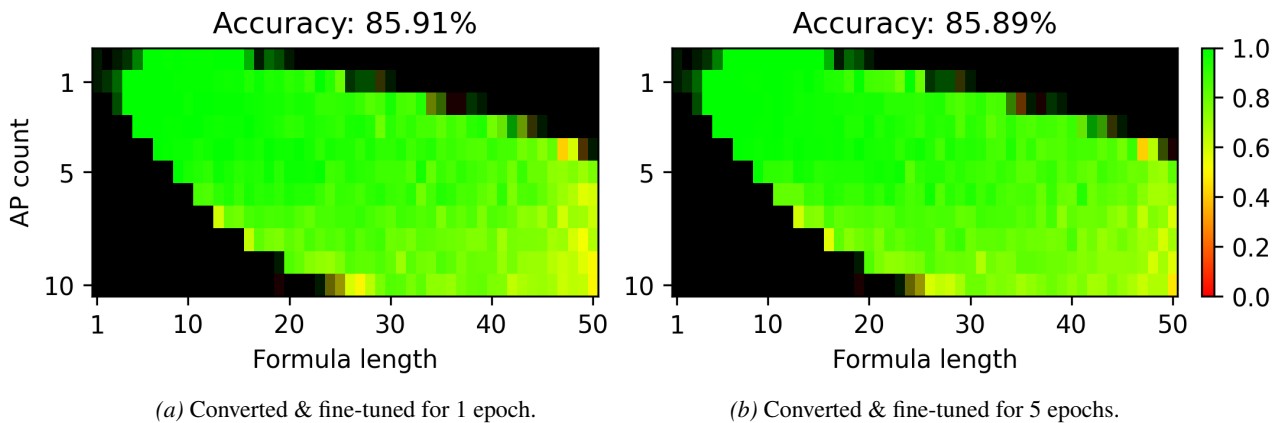

(a) Converted & fine-tuned for 1 epoch.
(b) Converted & fine-tuned for 5 epochs.

*Figure 11.* Heatmaps for the fine-tuned models on the LTL task. Both models are converted from the same pre-trained baseline. After one epoch of fine-tuning (a), the model already surpasses the from-scratch baseline (see Table 4). Five epochs of fine-tuning (b) yield comparable performance.

