# OpenReview forum: "Names Don’t Matter: Symbol-Invariant Transformer for Open-Vocabulary Learning"
_ICML.cc/2026/Conference — ICML 2026 regular_

### Official Review · Reviewer_hb4k · 2026-03-06

**Soundness:** 3
**Presentation:** 3
**Significance:** 2
**Originality:** 2
**Overall Recommendation:** 4
**Confidence:** 2

**Summary:**

They propose a novel Transformer-based mechanism that is provably invariant to the renaming of interchangeable tokens. They first create some parallel embedding streams where some interchangeable tokens are replaced with some placeholders. Each stream first goes through self-attention then they are aggregated and fed to cross-attention with original streams. Next, feed-forward networks are then applied to each stream. They theoretically prove the invariance to alpha-renaming and empirically validate the approach on open-vocabulary symbolic reasoning tasks, demonstrating generalization and state-of-the-art performance, including outperforming GPT-5.2 on LTL tasks.

**Compliance With Llm Reviewing Policy:**

Affirmed.

**Final Justification:**

Authors' reply fully address my concern for the paper

**Key Questions For Authors:**

1. is the method primarily intended for formal symbolic domains?


2. Would sufficiently large training data or sufficient task instruction prompt remove the need for architectural modifications?



3. For each stream do you use the same self-attention and MLP module?


4. Since there are multiple streams maintained during training and inference, how does memory usage scale with the number of interchangeable tokens? Could this become a bottleneck for longer sequences or larger models?

5. Do all streams share the same parameters for the self-attn and mlps, or are there any parameters specified for each stream?

**Limitations:**

yes

**Strengths And Weaknesses:**

Strengths:


The presentation is clear and easy to understand and also well-structured. Figure 1 helps readers clearly understand their proposed methods. The work does provide a deeper understanding, or highlight important properties of existing methods. The method is also distinguished from related literature and the novelty is well justified. I did not check the proof detail but the main idea and proof sketch is reasonable. The submission is technically sound and the claims are well supported. The ablation study is comprehensive and demonstrates the necessity of each component in the proposed method.






Weakness:


In table 1, you should also include the inference and training time for each model. As the method you proposed takes extra inference time and training time.


In Section 5.7 you mention ‘however, S is typically much smaller than L, and empirically, the method remains practical for S = 10: average time per sample increases from 3.38 ms to 5.13 ms on propositional logic (Figure 9 in Appendix J)’. When S is small, does alpha-renaming take comparable time as the method you proposed during training? And it may take less time during inference.


You compare your method with GPT5.2 on LTL tasks. I do not think it is a fair comparison. GPT5.2 may not be properly trained on LTL tasks but the model with your method is specifically trained on LTL. You can have an experiment that finetunes a pretrained LLMs on the same tasks and use the same amount of training compute to show your method is better.


As you scale the time complexity to $SL^2$, it is not applicable in regular text/language tasks. Why do we need to train a language model to solve this Symbol-Invariant LTL task? Can you prompt some coding LLMs to run some code to solve this task?


I think you identify one of the downsides of LLMs that they are symbol-variant but the proposed method is not quite applicable.

---

> ### Author Rebuttal · Authors · 2026-03-31
>
> We thank the reviewer for their careful reading and constructive feedback. We address each point below.
>
> ### Training/inference time comparisons
>
> We agree that including timing information in Table 1 would improve completeness. As reported in Section 5.7 and Figure 9, inference time scales linearly with $S$, increasing from 3.38 ms ($S=1$ baseline) to 5.13 ms ($S=10$) on propositional logic using a single NVIDIA L40S GPU. The alpha-renaming baseline uses a standard transformer with fixed embeddings and thus has essentially the same inference cost as the vanilla baseline. During training, our method processes $S$ streams per sample, so training time scales similarly to inference time with $S$.
>
> ### The fairness of the GPT-5.2 comparison
>
> GPT-5.2 benefits from massive pretraining that includes LTL content, since it's widely used in formal verification and well-represented in code and mathematics corpora. We also provided four few-shot examples covering varied formula complexities (Listings 1 and 2, Appendix H), and constrained outputs to JSON format via schema validation when possible. Despite these advantages, GPT-5.2 underperforms specialized models on LTL. This comparison aims to contextualize results against a strong general-purpose frontier model, not to make a like-for-like architectural claim.
>
> ### Q1: Is the method primarily intended for formal symbolic domains?
>
> Yes. We target domains where interchangeable tokens exist, e.g., formal logic, programming languages, theorem proving, graph reasoning with exchangeable node labels. We do not claim applicability to general natural language, where alpha-equivalence is not intrinsic. The $O(SL^2)$ cost is not critical in many formal domains because $S$ is typically small and bounded by the number of distinct symbolic variables.
>
> Also, as shown in our response to Reviewer ZfSC (Q2), our method can be applied to pre-trained models with some fine-tuning. Thus, we aim to explore coding and mathematical reasoning in future work.
>
> **Solver-based approaches** are an alternative, but neural sequence models offer complementary advantages: generalizing across formula structures, amortizing repeated solving over a training distribution, and generating approximate solutions far faster than exact solvers for PSPACE-complete problems like LTL (Section 5.4).
>
> ### Q2: Would sufficiently large training data or instruction prompts remove the need for architectural changes?
>
> No. A growing body of evidence shows that LLMs depend heavily on variable and identifier names, memorizing surface-level naming patterns rather than performing genuine semantic reasoning. Renaming variables (without altering semantics) causes significant performance degradation across code analysis tasks [1, 2, 3]. This cannot be remedied by scaling data or prompting, because the root cause is architectural: standard embedding tables assign distinct learned representations to distinct token identities, inherently encoding name-dependent biases. The theoretical guarantee our method provides (identical output for any alpha-equivalent input, by construction) is categorically impossible to achieve through data augmentation or prompting alone. We will cite [1, 2, 3] in the revision.
>
> Our renamed dataset perturbation experiment (Table 1) illustrates this: even the alpha-renaming baseline (explicit renaming augmentation) does not achieve perfect alpha-covariance and degrades under distribution shift. Our method achieves 100% alpha-covariance across all conditions by construction.
>
> ### Q3 & Q5: Parameter sharing
>
> All streams share identical parameters for self-attention, aggregated attention, and feed-forward networks. This is a core design choice that enables post-training vocabulary extension without retraining. We state this in Section 4.1 and will make it more prominent.
>
> ### Q4: Memory scaling
>
> Memory scales linearly with $S$, since each stream maintains a separate hidden state tensor of shape $L \times d$. On the LTL heatmap, the memory usage from 5 to 10 APs scales from 1.9 GB to 3.9 GB for the proposed method, and from 3.1 GB to 3.2 GB for the full-vocab baseline (for a batch size of 64). Note that the proposed model uses less parameters (Table 3) since we observed better parameter efficiency during preliminary hyperparameter tuning, which alleviates the memory scaling problem.
>
> For much larger $S$ or $d$, memory could become a bottleneck, and the sparsification strategies discussed with Reviewer kc6k (e.g., Top-K gating) would apply equally here. We will add a brief memory analysis alongside the time complexity discussion in Section 5.7.
>
> ### References
>
> * [1] Le et al. "When Names Disappear: Revealing What LLMs Actually Understand About Code."
> * [2] Miceli-Barone et al. "The Larger They Are, the Harder They Fail: Language Models Do Not Recognize Identifier Swaps in Python."
> * [3] Wang et al. "How Does Naming Affect LLMs on Code Analysis Tasks?"

---

> > ### Author Rebuttal · Reviewer_hb4k · 2026-04-05
> >
> > Thanks for your detailed reply! All my concerns have been solved

---

### Official Review · Reviewer_kc6k · 2026-03-12

**Soundness:** 3
**Presentation:** 4
**Significance:** 4
**Originality:** 4
**Overall Recommendation:** 5
**Confidence:** 3

**Summary:**

This paper addresses a fundamental limitation of standard Transformer architectures: their inability to structurally generalize to "interchangeable tokens" (e.g., bound variables, atomic propositions) in open-vocabulary settings. To solve this, the authors propose a novel "Symbol-Invariant Transformer." Instead of relying on statistical methods like random embeddings, the proposed architecture creates k parallel embedding streams for an input containing k interchangeable tokens. Using a combination of per-stream self-attention and an aggregated attention mechanism, the model isolates individual token contexts while sharing global information. Crucially, the authors mathematically prove that this architecture guarantees exact invariance to alpha-renaming (alpha-equivalence) by construction. The method is evaluated on synthetic copying tasks, propositional logic satisfiability, and Linear Temporal Logic (LTL) witness generation, demonstrating superior sample efficiency and out-of-distribution generalization compared to random-embedding baselines and GPT-5.2.

**Compliance With Llm Reviewing Policy:**

Affirmed.

**Key Questions For Authors:**

Complexity and Scaling: The complexity is O(SL2) where S is the number of interchangeable streams. While you show it works well for S≤10, real-world program synthesis or theorem proving might involve hundreds of local variables. Have you explored or theorized any sparsification or Top-K gating mechanisms to cap S during the forward pass without breaking the alpha-equivalence theorem?
Generating Unseen Symbols: In many formal reasoning tasks (e.g., constructive proofs, code generation), the model must generate a new variable name that was not present in the input encoder sequence. Can the current projection mechanism (Algorithm 5) handle the generation of a completely novel symbol? If not, please clarify this limitation.
GPT-5.2 Prompting: In your GPT-5.2 evaluation, were the models provided with few-shot examples in the prompt, or was it purely zero-shot? Formal logic generators often suffer heavily from formatting errors in zero-shot settings.
The k=0 Edge Case: In Appendix A, you mention that if an input contains no interchangeable tokens, k=1 and the model cannot generate interchangeable tokens. Does this cause practical training instability, or require a separate fallback routing in the code?

**Limitations:**

While the authors commendably and honestly discuss the O(SL2 ) computational complexity in Section 5.7, they should add a dedicated "Limitations" section to address two additional points:The title's claim of "Open-Vocabulary" is practically bounded by the hardware limits on S (number of parallel streams).
The apparent structural limitation regarding the model's inability to generate completely novel tokens in the decoder that were never instantiated as streams in the input sequence. Softening these claims will make the paper more scientifically rigorous.

**Strengths And Weaknesses:**

Soundness：
Strengths: The technical foundation of this paper is exceptionally strong. Theorem 4.1, which guarantees alpha-renaming invariance by exploiting the commutativity of the aggregation operations across parallel streams, is elegant and rigorous. The experimental design is very thorough: the authors use dataset perturbations (renamed and reduced datasets) to perfectly isolate and prove the value of the model's inductive bias. Furthermore, the ablation studies (Appendix G) are exhaustive and clearly illustrate the necessity of the proposed per-stream and aggregated attention mechanisms across different reasoning tasks.
Weaknesses: The comparison against GPT-5.2 is somewhat an "apples-to-oranges" comparison. Comparing a 3M-parameter, task-specifically trained expert model that outputs constrained JSON/syntax against a general-purpose LLM via prompting is not entirely fair, especially since LLMs often fail formal logic tasks due to minor syntactic deviations rather than pure reasoning failures. Additionally, the paper does not sufficiently address whether this architecture can generate a completely novel symbol in the output that did not appear in the input sequence.

Presentation：
Strengths: The paper is exceptionally well-written and easy to follow. Figure 1 provides a crystal-clear visual explanation of the rather complex multi-stream architecture. The pseudocode in Appendix A ensures reproducibility. The authors are also intellectually honest in Section 5.7, openly discussing the
O(SL 2) computational complexity tradeoff.
Weaknesses: The phrase "Open-Vocabulary Learning" in the title and abstract is slightly misleading. Because the computational complexity scales linearly with the number of interchangeable tokens (S), the vocabulary is practically "bounded" by memory and compute. "Dynamic Vocabulary Extension" or "Bounded Open-Vocabulary" might be more accurate descriptions.

Significance：
Strengths: This paper presents a major paradigm shift for neuro-symbolic AI. Moving from statistical approximations (like random embeddings) to structural, mathematical guarantees for alpha-equivalence is a highly significant contribution that will likely influence future architectures in theorem proving, formal verification, and program synthesis.
Weaknesses: The current empirical evaluation is restricted to purely synthetic, highly structured logic tasks. Demonstrating this architecture on a more practical task (e.g., code generation with variable renaming, or Lean theorem proving) would significantly elevate the paper's impact, though the current scope is sufficient to prove the core concept.

Originality：
Strengths: While permutation-invariant neural networks (e.g., Set Transformers) exist, creating parallel embedding streams with "actual" and "placeholder" indices, combined with the specific asymmetric projection and aggregation mechanisms, is a highly novel and creative solution specifically tailored to the problem of interchangeable tokens in sequence-to-sequence models.

---

> ### Author Rebuttal · Authors · 2026-03-31
>
> We thank the reviewer for the thorough and generous assessment. We address each question below.
>
> ### Q1: Scaling and sparsification for large S.
>
> We agree this is an important direction. In the current work, S is bounded by the number of distinct interchangeable tokens in the input, which is small for the tasks studied (up to 10 APs). For domains with hundreds of variables, the $O(SL^2)$ cost would indeed become significant. While we have not yet implemented sparsification, several promising directions exist. Alongside top-K gating suggested by the reviewer, examples include low-rank attention approximations applied per-stream and adaptive stream selection, which were briefly mentioned in the conclusion section as future directions.
>
> Importantly, any such reduction must be designed carefully to preserve the alpha-equivalence guarantee: a top-K approach that drops streams based on token identity would break invariance, whereas one that drops based on input-symmetric criteria (e.g., positional frequency) could potentially preserve it. We will update our paper to include this discussion, complementing the brief mention in Section 6.
>
> For more information, please feel free to read our response to Reviewer hb4k, where we discuss memory scaling.
>
> ### Q2: Generating completely novel symbols not present in the input.
>
> This is an insightful question that points to an exciting future direction.
>
> The proposed method can generate tokens that don't appear in the input, similarly to conventional language models. However, any symbol that should be treated as an interchangeable token in the output must have a corresponding embedding stream. Since we currently instantiate the streams based on the encoder input, the model cannot produce an interchangeable token if it doesn't appear in the input. For the tasks studied (satisfying assignment prediction, LTL witness generation), output interchangeable tokens are always drawn from the input formula, so this does not affect our benchmarks.
>
> Extending the method to support fresh interchangeable tokens not present in the input (as required in constructive theorem proving or code generation where new variable names must be invented) is an exciting direction for future work. A straightforward approach would be to maintain a pool of reserved "fresh symbol" streams. We will add this discussion to the paper.
>
> ### Q3: GPT-5.2 prompting setup.
>
> The prompts used for GPT-5.2 include four few-shot examples for both tasks, as shown in Listings 1 and 2 in Appendix H. The examples cover a range of formula complexities and demonstrate the expected input/output format. For the propositional logic task, outputs were additionally constrained to JSON format via schema validation to minimize formatting errors. We will make the few-shot nature of the prompting more prominent in the main text, as it is currently only visible in the appendix.
>
> ### Q4: The $k=0$ edge case and training stability.
>
> As described in Appendix A, when the input contains no interchangeable tokens we create a single stream (k=1) that does not correspond to any specific interchangeable token. In this mode, per-stream and aggregated attention collapse to the same operation since there is only one stream, so no special routing logic is required in practice. We did not observe any training instability from this edge case in our experiments: samples with zero interchangeable tokens are relatively rare in the logic datasets (as shown in Figure 3, there are very few 0-AP samples), and the model handles them gracefully as a single-stream case. No separate fallback branch is needed in the implementation.
>
> ### On the "Open-Vocabulary" framing.
>
> We thank the reviewer for their thoughtful remark on terminology. In this work, we use "open-vocabulary" to describe a model’s ability to reason about concepts that were not explicitly observed during training, rather than being limited to a fixed, predefined set of labels (i.e., a closed vocabulary). This usage aligns with prior work in the open-vocabulary learning literature, as well as the capability our method provides: the model generalizes to interchangeable token sets strictly larger than those seen during training, without retraining.
>
> As the reviewer recommends, we will add a dedicated limitations section to the revision and discuss the practical limits to the open-vocabulary capability, as well as the points raised in Q2.

---

> > ### Author Rebuttal · Reviewer_kc6k · 2026-04-05
> >
> > Thank you for the author's reply.

---

### Official Review · Reviewer_ZfSC · 2026-03-24

**Soundness:** 3
**Presentation:** 3
**Significance:** 2
**Originality:** 3
**Overall Recommendation:** 4
**Confidence:** 2

**Summary:**

The paper proposes a new transformer mechanism to deal with the model's sensitivity to renaming of interchangeable tokens in symbolic reasoning. The method utilizes parallel streams for each individual interchangeable tokens combined with an aggregation mechanism to guarantee invariance. Experiments on propositional logic and linear temperal logic demonstrate the effectiveness of the approach.

**Compliance With Llm Reviewing Policy:**

Affirmed.

**Key Questions For Authors:**

1. Can the proposed mechanism be applied to decoder-only autoregressive models?
2. How does your method benefit pre-trained LLMs on more realistic tasks like coding and math reasoning?
3. How do you determine the set of interchangeable tokens and the number of streams in practice?

**Limitations:**

Yes

**Strengths And Weaknesses:**

Overall, the paper addresses an important limitation of transformers, namely their sensitivity to renaming of interchangeable tokens. The effectiveness of the proposed mechanism is guaranteed by construction, and further supported by strong empirical results. The paper is clearly written and easy to follow.

One major concern I have about the proposed method is its applicability to real-world scenarios. The experiments are conducted on an encoder-decoder transformer model trained on a dedicated training set, and only tested on tasks with a clearly defined variable set. It remains unclear whether the proposed method can be effectively applied to a pre-trained autoregressive LLM on real-world tasks like math reasoning and coding. Strategies to identify the set of interchangeable tokens in practice are also not discussed.

---

> ### Author Rebuttal · Authors · 2026-03-31
>
> We thank the reviewer for their positive assessment and constructive feedback. We address each concern below.
>
> ### Q1: Decoder-only models.
>
> Our method is presented for encoder-decoder models, but the core mechanism (parallel embedding streams with per-stream and aggregated attention) is applicable to decoder-only models as well. The self-attention in our architecture is structurally analogous to what appears in a decoder-only transformer; the main adaptation required is applying causal masking throughout, which we already do in our decoder. We will clarify this in the revision.
>
> ### Q2: Applicability to pre-trained models and realistic tasks.
>
> To show that our method can be applied to pre-trained models, we performed an additional experiment. If all aggregated attention components are disabled, the model does not introduce any new parameters compared to the baseline. We propose the following process to convert a pre-trained baseline model to our method:
> 1. Use "a" (one of the interchangeable tokens) embedding as the actual embedding (Section 4.1).
> 2. Use "b" (another interchangeable token) as the placeholder embedding.
> 3. All other learnable parameters map 1-to-1. Fine-tune the new model.
>
> We present the results for the LTL task below. The first row is taken from Table 4 (EP-DP-CP model). Pre-trained baseline refers to the model with fixed embeddings, which was trained on 5 APs and is unable to process larger vocabularies. We convert this model to our method and fine-tune it.
>
> | Model | 5 AP Val | 10 AP Val | Heatmap (avg.) |
> | ----- | -------- | --------- | ------- |
> | Same arch, trained from scratch | 94.77% | 91.34% | 84.13% |
> | Pre-trained baseline model | 97.96% | N/A | N/A |
> | Converted & fine-tuned (1 epoch) | 94.75% | 92.44% | 85.91% |
> | Converted & fine-tuned (5 epochs) | 95.12% | 92.02% | 85.88% |
>
> The fine-tuned models achieve very similar performance to the models trained from scratch, improving the applicability of our method to pre-trained models. Using this process, a pre-trained model with fixed embeddings can be transformed to our method to gain theoretical alpha-invariance guarantees.
>
> On the question of realistic tasks like coding and math reasoning: we agree this is a valuable direction. We note, however, that the tasks studied here (LTL solving and propositional logic) are non-trivial symbolic reasoning problems with direct relevance to formal verification and program analysis. The paper explicitly mentions program analysis and theorem proving as future application domains (Section 6). Demonstrating the approach on these domains would require domain-specific infrastructure (e.g., formal verifiers for correctness checking) that is beyond the current scope, but the fine-tuning result above is an encouraging signal for transfer to pre-trained models.
>
> ### Q3: Determining the set of interchangeable tokens and number of streams.
>
> The set of interchangeable tokens is determined by the task domain and is typically straightforward to identify: in logic, they are the atomic propositions; in programming languages, they are the bound variables; in theorem proving, they are the universally quantified variables. This is analogous to how one identifies special token types in standard NLP preprocessing. We will add a discussion of this to the paper.
>
> The number of streams S is set equal to the number of distinct interchangeable tokens present in the input at inference time. Since all streams share parameters, adding more streams at test time (for vocabulary generalization) incurs no retraining, only a linear increase in computation, which we show remains practical up to S = 10 (average inference time of 5.13 ms vs. 3.38 ms for the baseline, as shown in Figure 9). We will make this more explicit in the paper.
>
> You may also be interested in our response to Reviewer hb4k for discussion on memory scaling.

---

> > ### Author Rebuttal · Reviewer_ZfSC · 2026-04-05
> >
> > Thank you for your response. I was hoping to see experiments that demonstrate the effectiveness of applying the proposed method to modern decoder-only LLMs, which would significantly strengthen its practical value. I still lean towards acceptance so I will keep my score.

---

### Decision · Program_Chairs · 2026-04-30

**Decision:**

Accept (regular)

**Comment:**

In formal domains (code, mathematics), variable names do not carry meaning. Standard Transformers do not respect this: they assign a fixed embedding to each symbol and renaming variables leads to different predictions. This paper proposes a novel architecture that processes placeholder symbols through a parallel stream, then aggregates information across streams. This design guarantees that the model's output is unchanged when symbols are renamed. The proposed architecture is evaluated on propositional logic satisfiability and temporal logic witness generation, where it outperforms baselines and achieves perfect invariance to renaming.

All reviewers agree the paper is well-written. The invariance guarantee is an interesting theoretical result, and the method can be integrated into standard encoder-decoder Transformers as a lightweight modification. Empirical results are convincing, with perfect invariance across all settings. The ablation study is thorough and informative.

On the other hand, the practical value of the contribution is limited:
1) the evaluation is narrow, limited to two synthetic logic tasks with up to 10 symbols,
2) the path to a real-world use is unclear (no experiments involve pretrained LLMs, decoder-only architectures, or tasks like code analysis or theorem proving),
3) the "open-vocabulary" is actually bounded by the number of parallel streams.

The interesting solution to a well motivated problem warrants acceptance in my opinion.